# Naked mole-rat brown fat thermogenesis is diminished during hypoxia through a rapid decrease in UCP1

Hang Cheng [1], Rajaa Sebaa[2,3,4], Nikita Malholtra[1], Baptiste Lacoste [5,6,7], Ziyad El Hankouri [2,3], Alexia Kirby[1], Nigel C. Bennett[8], Barry van Jaarsveld[8], Daniel W. Hart [8], Glenn J. Tattersall [9], Mary-Ellen Harper [2,3,10 ✉] & Matthew E. Pamenter [1,6,10 ✉]

Naked mole-rats are among the most hypoxia-tolerant mammals. During hypoxia, their body temperature ($T_b$) decreases via unknown mechanisms to conserve energy. In small mammals, non-shivering thermogenesis in brown adipose tissue (BAT) is critical to $T_b$ regulation; therefore, we hypothesize that hypoxia decreases naked mole-rat BAT thermogenesis. To test this, we measure changes in $T_b$ during normoxia and hypoxia (7% $O_2$; 1–3 h). We report that interscapular thermogenesis is high in normoxia but ceases during hypoxia, and $T_b$ decreases. Furthermore, in BAT from animals treated in hypoxia, UCP1 and mitochondrial complexes I-V protein expression rapidly decrease, while mitochondria undergo fission, and apoptosis and mitophagy are inhibited. Finally, UCP1 expression decreases in hypoxia in three other social African mole-rat species, but not a solitary species. These findings suggest that the ability to rapidly down-regulate thermogenesis to conserve oxygen in hypoxia may have evolved preferentially in social species.

[1] Department of Biology, University of Ottawa, Ottawa, ON, Canada. [2] Department of Biochemistry, Microbiology, and Immunology, Faculty of Medicine, University of Ottawa, Ottawa, ON, Canada. [3] Ottawa Institute of Systems Biology, University of Ottawa, Ottawa, ON, Canada. [4] Department of Medical Laboratories, College of Applied Medical Sciences, University of Shaqra, Duwadimi, Saudi Arabia. [5] Department of Cellular and Molecular Medicine, Faculty of Medicine, University of Ottawa, Ottawa, ON, Canada. [6] University of Ottawa Brain and Mind Research Institute, Ottawa, ON, Canada. [7] Neuroscience Program, Ottawa Hospital Research Institute, Ottawa, ON, Canada. [8] Department of Zoology and Entomology, University of Pretoria, Pretoria, South Africa. [9] Department of Biological Sciences, Brock University, St. Catharines, ON, Canada. [10] These authors contributed equally: Mary-Ellen Harper, Matthew E. Pamenter. ✉email: mharper@uottawa.ca; mpamenter@uottawa.ca

Animals that inhabit hypoxic environments have evolved elegant suites of physiological adaptations that enable them to thrive in low-oxygen niches[1,2]. The key to tolerating hypoxia is to match metabolic demand to reduced oxygen supply[3–5], and hypoxia-tolerant animals typically exhibit robust decreases in metabolic rate when oxygen supply is limited[1,6]. Cold-induced thermogenesis is among the most energy-intensive processes in small mammals and many hypoxia-tolerant mammals employ thermoregulatory strategies to reduce body temperature ($T_b$) and facilitate reduced metabolic demand in hypoxia[7,8]. Naked mole-rats (*Heterocephalus glaber*) are among the most hypoxia-tolerant mammals identified and tolerate minutes of complete anoxia, hours at 3% $O_2$, and days to weeks at 8–10% $O_2$[9–11]. In acute hypoxia or anoxia, the rate of oxygen consumption (an indirect measure of metabolic rate) of adult naked mole-rats decreases by up to 85%[12,13], and $T_b$ decreases to near-ambient levels[14]. Although this degree of metabolic rate and $T_b$ suppression is not remarkable among hypoxia-tolerant species[6], it is important to note that other mammals that are capable of similar or more extreme metabolic rate suppression in severe hypoxia typically enter into a coma- or torpor-like state until $O_2$ levels are restored[6,15]. Conversely, naked mole-rats remain conscious and active in hypoxia, albeit to a reduced degree[14].

Naked mole-rats meet the definition of heterotherms because they are weakly capable of thermogenesis but are generally not very efficient thermoregulators[14,16]. This is primarily due to their lack of insulating fur and fat[17], which allows for rapid loss of heat through convective mechanisms. Indeed, naked mole-rats expend considerable energy to thermoregulate outside of their thermoneutral zone and several studies have reported their ability to maintain $T_b$ well above $T_a$ in a range of temperatures ($T_b-T_a$ differential ranging from 0.0 to >13.0 °C in animals exposed to $T_a$'s ranging from 37 to 10 °C)[18–20]. Concomitantly, metabolic rate is substantially elevated in cold temperatures, likely reflecting the high metabolic cost of thermogenesis[14,18,21]. This ability to thermoregulate, even weakly, is likely important to the ecophysiology of this species as naked mole-rat burrow temperatures have a warm but variable thermal range (<25 °C to <45 °C[22]).

Importantly, the metabolic cost that may be attributed to thermoregulation in the cold is substantial, and the metabolic rate of naked mole-rats held in cold temperatures is >2- to 4-fold greater than when they are held within their thermoneutral zone[14,18]. However, the whole-animal metabolic rate of naked mole-rats in acute hypoxia is not significantly different between cold and warm experimental conditions[14]. These observations suggest two intriguing hypotheses: specifically, that naked mole-rats: (1) employ active thermogenesis in cold, normoxic conditions, and (2) thermogenesis is reduced during acute hypoxia to conserve oxygen and support the robust hypoxic metabolic rate suppression previously reported in this species[9,12,14].

The primary mechanism of cold-induced thermogenesis in small mammals is uncoupling protein-1 (UCP1)-mediated mitochondrial uncoupling in brown adipose tissue (BAT) (i.e., non-shivering thermogenesis (NST)). Activation of UCP1 uncouples mitochondrial respiration from ATP-synthesis, resulting in heat generation through futile cycling of the electron transport chain (ETC)[23,24]. Importantly, naked mole-rats express functional BAT[16]; however, no study has directly examined thermogenic adaptations to hypoxia or the underlying mechanisms in this species. Given the high cost of thermoregulation (particularly for a small and naked rodent) and the requisite need to lower metabolic demand in hypoxia, such an investigation is of pressing interest in the study of ecophysiological adaptations to life in hypoxia in this fascinating species. Therefore, in the present study we employ in vivo thermal imaging to interrogate thermogenic adaptations to acute hypoxia in naked mole-rat interscapular BAT (iBAT). We also explore the underlying molecular mechanism regulating hypoxic changes in iBAT function from naked mole-rats and several closely related and hypoxia-tolerant African mole-rat species.

## Results

We first examined surface temperatures in awake and freely behaving naked mole-rats during 1 h of normoxia and at 30 °C, which is the $T_a$ at which our animals are housed. We observed evidence for robust heat production in normoxia, with a particularly high degree of heat production apparent in the interscapular region, and that the $T_b$ of naked mole-rats was ~3 °C > $T_a$ (Fig. 1A, C; $n = 11$ independent animals). Upon exposure to an acute hypoxic challenge (7% $O_2$) surface temperature rapidly fell such that the animal largely disappeared into the background in the thermal image (Fig. 1B, C and Supplementary Movie 1). In all experiments, the temperature measured from the interscapular region was greater than the dorsal surface skin temperature or the core $T_b$ in normoxia, but with the onset of hypoxia, all body temperature measurements (including implanted radiofrequency identification (RFID) chip and forward-looking infrared (FLIR) analysis), including in the interscapular region, collapsed to within 1 °C of the $T_a$ (Fig. 1D; $F_{2,30} = 136.0$, $p < 0.0001$). Upon reoxygenation, all temperature measurements returned towards normoxic values following a lag-time of ~ 20–40 min ($p < 0.0001$).

We then repeated this experimental protocol at an $T_a$ of 20 °C to increase the thermal scope within which animals might respond to a hypoxic challenge (Fig. 2A, B; $n = 8$ independent animals). In this colder temperature and under normoxia, naked mole-rat $T_b$ was ~ 7–9 °C > $T_a$, suggesting that these animals were actively attempting to thermoregulate in the cold $T_a$. However, with the onset of hypoxia, all body temperature measurements again collapsed towards the $T_a$, such that $T_b$ in hypoxia was ~ 1 °C > $T_a$ ($F_{2,15} = 29.89$, $p < 0.0001$). Finally, we repeated this experimental protocol at an $T_a$ above the thermoneutral zone of naked mole-rats (36 °C, Fig. 2C, D; $n = 12$ independent animals). At this $T_a$, $T_b$ was not significantly different from $T_a$ and the degree of thermogenesis, as measured by interscapular thermal imaging, was minimal. However, a small but significant decrease in both $T_{BAT}$ and $T_{rump}$ occurred with the onset of hypoxia ($F_{2,22} = 6.63$, $p = 0.0113$). With the onset of hypoxia and subsequent reoxygenation, $T_b$ was unchanged ($p = 0.0925$ and 0.4972, respectively).

In response to cold exposure, the mammalian sympathetic nervous system releases noradrenaline to stimulate β-adrenergic receptors, which activates lipolysis in white adipose tissue (WAT) and BAT to support the high energetic costs of UCP1-driven thermogenesis[25]. These properties of BAT and UCP1 led us to reinvestigate the ability of naked mole-rats to regulate their $T_b$ through BAT thermogenesis upon adrenergic activation in vivo. Specifically, we examined the regulation of BAT NST via the injection of saline (sham controls) or the general β-adrenergic agonist isoproterenol (dissolved in saline). Saline alone had no effect on $T_b$ or thermogenesis in normoxia or hypoxia (Fig. 3A, B; $n = 5$ independent animals); however, injection of isoproterenol resulted in enhanced thermogenesis in the interscapular region such that surface temperatures from this region were ~2–3 °C greater than in sham-treated controls (Fig. 3C, D and Supplementary Movie 2; $F_{2,30} = 41.2$, $p < 0.0001$, $n = 11$ independent animals). Intriguingly, with the subsequent onset of hypoxia, this enhanced activation of interscapular thermogenesis was nonetheless curtailed and all measurements of $T_b$ decreased to within ~1 °C of the $T_a$ (30 °C).

To understand the mechanism underlying the rapid change in interscapular thermogenesis, we next held naked mole-rats at

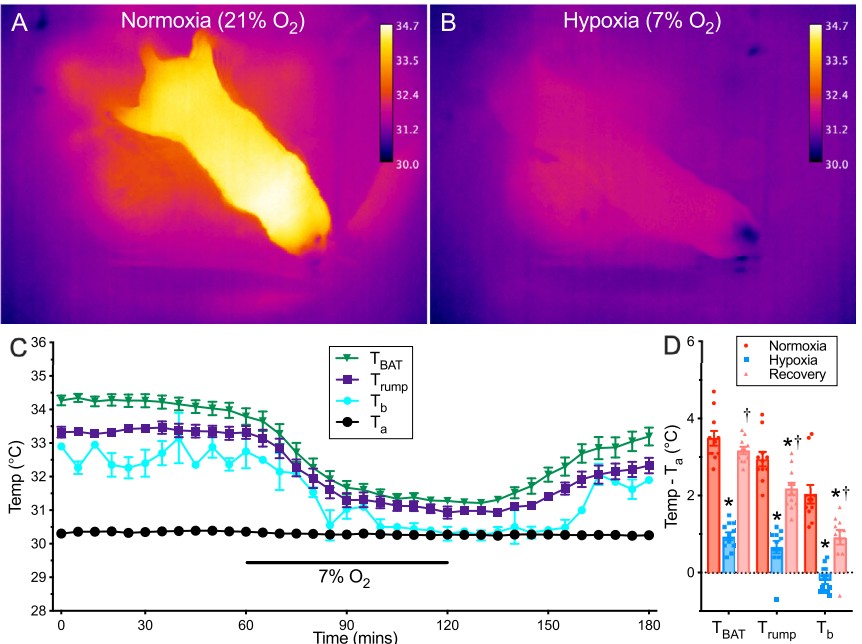

**Fig. 1 Thermogenesis ceases in acute hypoxia and body temperature drops to ambient levels. A, B** FLIR thermal images of a naked mole-rat following 60 min of exposure to normoxia (**A**, 21% $O_2$) or hypoxia (**B**, 7% $O_2$). **C** Summaries of ambient temperature ($T_a$, black circles), core body temperature ($T_b$, teal circles), interscapular brown adipose tissue temperature ($T_{BAT}$, green triangles), and dorsal skin surface temperature ($T_{rump}$, blue squares) from naked mole-rats exposed to a normoxia → hypoxia → recovery protocol in 30 °C (n = 11 independent animals). **D** Summaries of temperature difference between physiological temperatures and $T_a$ in the final 10 min of each treatment period from (**C**). Data are mean ± SEM. Asterisks indicate significant difference from normoxic controls; daggers indicate significant difference from hypoxia (repeated measures ANOVA with Tukey post-test; $F_{2,30} = 136.0$, $p < 0.0001$). Source data are provided as a Source Data file.

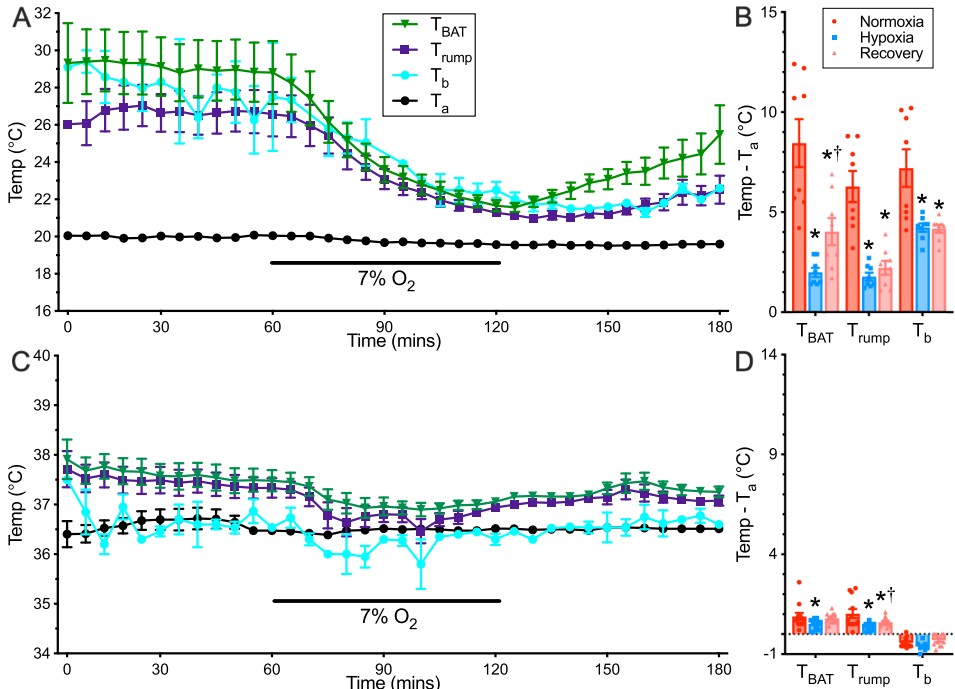

**Fig. 2 Naked mole-rats actively thermoregulate in cold, but not hot temperatures. A, C** Summaries of ambient temperature ($T_a$, black circles), core body temperature ($T_b$, teal circles), interscapular brown adipose tissue temperature ($T_{BAT}$, green triangles), and dorsal skin surface temperature ($T_{rump}$, blue squares) from naked mole-rats exposed to a normoxia → hypoxia → recovery protocol in 20 °C (**A**, n = 8 independent animals), or 36 °C (**C**, n = 12 independent animals). **B, D** Summaries of temperature difference between physiological temperatures and $T_a$ at 20 °C (**B**) or 36 °C (**D**) from data presented in (**A**) and (**C**), respectively. Data are mean ± SEM. Asterisks indicate a significant difference from normoxic controls; daggers indicate significant difference from hypoxia (repeated measures ANOVA with Tukey post-test; $F_{2,15} = 29.89$, $p < 0.0001$ for 20 °C, $F_{2,22} = 6.63$, $p = 0.0113$ for 30 °C). Source data are provided as a Source Data file.

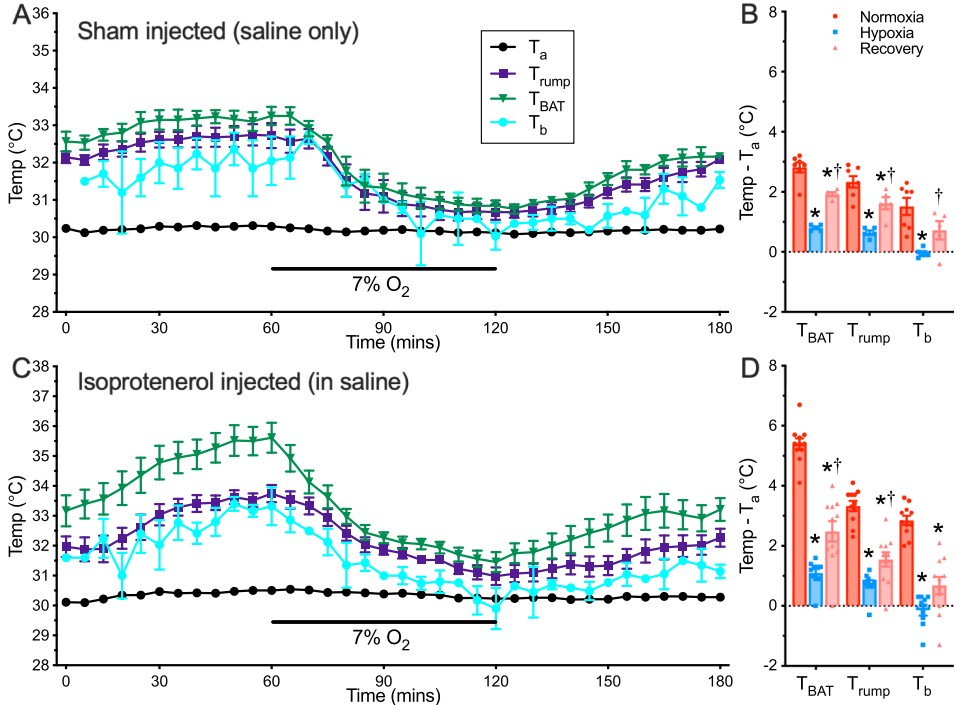

**Fig. 3 Adrenergic stimulation enhances non-shivering thermogenesis in normoxia, but not in hypoxia. A**, **C** Summaries of ambient temperature ($T_a$, black circles), core body temperature ($T_b$, teal circles), interscapular brown adipose tissue temperature ($T_{BAT}$, green triangles), and dorsal skin surface temperature ($T_{rump}$, blue squares) from naked mole-rats exposed to a normoxia → hypoxia → recovery protocol in 30 °C following injection of saline (**A**, $n = 5$ independent animals), or isoproterenol (**C**, $n = 11$ independent animals). **B**, **D** Summaries of temperature difference between physiological temperatures and $T_a$ for the same animals as in (**A**, **C**). Data are mean ± SEM. Asterisks indicate significant difference from normoxic controls; daggers indicate a significant difference from hypoxia (repeated measures ANOVA with Tukey post-test; $F_{4,9} = 0.5272$, $p < 0.7190$ for shams, $F_{2,30} = 41.2$, $p < 0.0001$ for isoproterenol). Source data are provided as a Source Data file.

normoxia or for 1 or 3 h in acute hypoxia (7% $O_2$) at 30 °C, and then rapidly dissected iBAT. We then examined the expression of thermogenic UCP1 and UCP3 proteins as well as that of mitochondrial oxidative phosphorylation (OXPHOS) complexes I–V. We found that, following 1 and 3 h of acute hypoxia, UCP1 protein expression decreased by 58–62% relative to normoxic controls (Fig. 4A, B; $F_{2,33} = 10.96$, $p = 0.0002$, $n = 19$, 11, and 6 biologically independent replicates for normoxia and 1 and 3 h hypoxia, respectively). UCP3 protein expression tended to decrease with progressive hypoxia, but this change did not reach significance (Fig. 4A, B; $F_{2,15} = 2.06$, $p = 0.162$, $n = 6$ biologically independent replicates each). We also analyzed the number of UCP1-expressing cells in iBAT samples using immunohistochemical approaches and found that the percent of cells expressing UCP1 decreased by ~9% at 3 h of hypoxia but was unchanged after 1 h of hypoxia (Supplementary Fig. S1A–D; $F_{2,19} = 3.684$, $p = 0.0442$, $n = 12$, 11, and 11 biologically independent replicates for normoxia and 1 and 3 h hypoxia, respectively). In addition, the expression of mitochondrial ETC complexes I–IV proteins decreased by 37–63% in 1 and 3 h hypoxia (Fig. 4C, D; $F_{4,165} = 60.90$, $p < 0.0001$, $n = 19$, 11, and 6 biologically independent replicates for normoxia and 1 and 3 h hypoxia, respectively), while the expression of the $F_1F_o$-ATPase (complex V) decreased after 3 h ($p = 0.0032$), but not 1 h of hypoxia exposure ($p = 0.6012$; Fig. 4C, D). We also examined lipid droplet (LD) area within the cytoplasm of naked mole-rat iBAT samples, but the variability in this analysis was large and no significant changes were observed (Supplementary Fig. S1E–H; $F_{2,31} = 0.85$, $p = 0.1461$; $n = 12$, 11, and 11 biologically independent replicates for normoxia and 1 and 3 h hypoxia, respectively).

Our observation of rapid changes in iBAT mitochondrial protein expression prompted further investigation of the underlying mechanisms. Since UCP1, the ETC complexes, and the $F_1F_o$-ATPase are all situated within the mitochondrial inner membrane[26,27], we next examined changes in mitochondrial ultrastructure and cristae morphology in iBAT from naked mole-rats exposed to hypoxia in vivo. Using immunoblotting, we measured changes in the expression of TOM20, which is located in the mitochondrial outer membrane, as a marker of mitochondrial content. Expression of TOM20 exhibited a decreasing trend with progressive hypoxia and was significantly reduced by 3 h of hypoxia (Fig. 5A, B; $F_{2,15} = 3.98$, $p = 0.0428$; $n = 6$ biologically independent replicates each).

This change in TOM20 protein expression prompted further investigations in mitochondrial ultrastructure. We next used transmission electron microscopy (TEM) to examine iBAT mitochondria from naked mole-rats treated in normoxia, 1 or 4 h hypoxia, or 4 h of hypoxia with 1 h of recovery in normoxia. We found that iBAT from normoxic naked mole-rats contained abundant mitochondria with dense cristae and intact inner membranes. Conversely, mitochondria in iBAT from hypoxic naked mole-rats displayed significant morphological alterations. After 1 h of hypoxia, total mitochondrial area was significantly decreased compared with normoxic conditions, consistent with the observed decrease in TOM20; and subsets of mitochondria were more tortuous and had sparser cristae (Fig. 5D). Following 4 h of hypoxia, we measured a significant decrease in cristae length and number (Fig. 5E). After recovery, all structural indices were comparable to normoxic conditions, except average cristae number which failed to fully recover (Fig. 5). There were also degradative, lysosome-like vacuoles between iBAT mitochondria from hypoxic naked mole-rats (Fig. 5D, red arrow). Next, we performed blinded qualitative analysis of mitochondria ultrastructure and cristae morphology ($n = 22$–42 technical replicates from four biological replicates per treatment; Fig. 5G–J). We

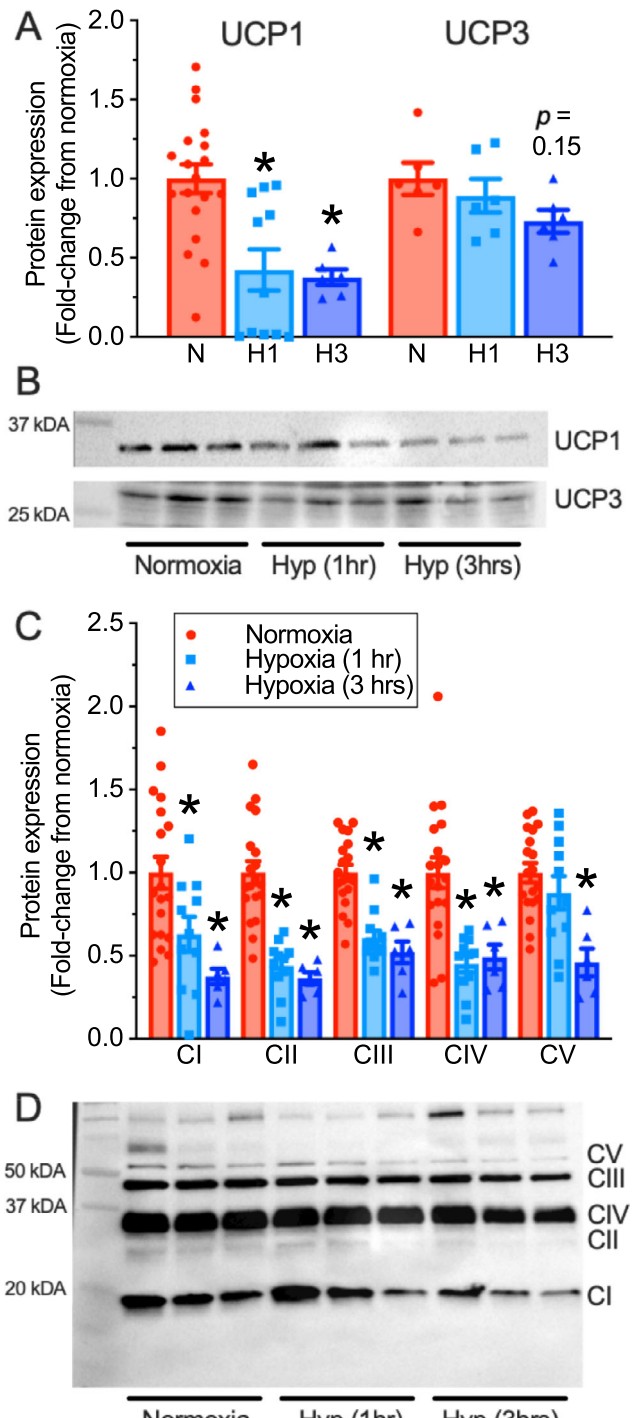

**Fig. 4 Thermogenic and oxidative phosphorylation protein expression decreases in acute hypoxia. A** Summary of uncoupling protein-1 (UCP1) and uncoupling protein-3 (UCP3) expression in interscapular brown adipose tissue (iBAT) from naked mole-rats treated in normoxia (21% $O_2$, red bars; $n = 18$ and 6 biologically independent samples for UCP1 and UCP3, respectively), or 1 or 3 h of hypoxia (7% $O_2$, light and dark blue bars, respectively; $n = 11$ and 6 biologically independent samples for UCP1 and 6 for each time point for UCP3). **B** Western blot images of UCP1 and UCP3 protein bands from (**A**). **C** Summary of electron transport chain (ETC) complexes I–IV (CI–CIV) and $F_1F_o$-ATPase (CV) protein expression in iBAT from naked mole-rats treated as in (**A**). **D** Western blot image of ETC complexes I–IV and $F_1F_o$-ATPase protein bands. Data are mean ± SEM. Asterisks indicate significant difference from normoxic controls (one-way ANOVA with Dunnett's multiple comparisons test; $F_{2,33} = 10.96$, $p = 0.0002$ for UCP1; $F_{4,165} = 60.90$, $p < 0.0001$ for ETC proteins; $p > 0.05$ for other proteins). Additional abbreviations: N – normoxia, H1 – 1 h hypoxia, H3 – 3 h hypoxia. Source data are provided as a Source Data file.

Fig. S2). We then examined co-immunoprecipitation between UCP1 and ubiquitin, but did not find any relation. As we could not find a direct link between UCP1 and ubiquitin, we further investigated other potential protein degradation mechanisms, such as mitophagy. Specifically, we next used immunoblotting to explore changes in key activators of mitophagy, including parkin, PINK1, p62, FUNDC1, Nix, BNIP3, TBC1D15, RAB7A, and LC3-I and II proteins in iBAT following 1 or 3 h of in vivo hypoxia. There were no increases in the expression of any of these proteins in hypoxia, except for LC3-II, which increased ~50% after 3 h of hypoxia ($F_{2,12} = 4.14$, $p = 0.0441$; Fig. 6; $n = 6–12$ biologically independent replicates for each treatment condition). In addition, the expression of BNIP3 decreased 42–53% with progressive hypoxia ($F_{2,15} = 6.16$, $p = 0.0111$). This change may be related to the role of this protein in regulating apoptosis, instead of in mitophagy (see below)[29].

Changes in mitochondrial size and shape are often mediated by mitochondrial fission or fusion and so we next examined the expression of key mediators of these processes in naked mole-rat iBAT (Fig. 7; $n = 6-12$ biologically independent replicates each). We found that progressive hypoxia increased phosphorylation of DRP1 at serine 616 (DRP-S616; ~2.25-fold increase; $F_{2,15} = 4.492$, $p = 0.0296$) and upregulated the expression of FIS1 by 243% at 3 h of hypoxia ($F_{2,33} = 23.83$, $p < 0.0001$), indicating the occurrence of fission in iBAT mitochondria with acute in vivo hypoxia. Conversely, canonical mediators of fusion, including OPA1, and MFN1 and MFN2 were unchanged.

Cellular apoptosis can occur in conjunction with mitophagy and autophagy and/or mitochondrial fission events and so we also examined activation of apoptosis pathways (Fig. 8; $n = 6$ biologically independent replicates each). Intriguingly, we found that key activators of apoptosis were downregulated during acute in vivo hypoxia, including p53 (by 38% and 63% at 1 and 3 h hypoxia, respectively; $F_{2,15} = 6.087$, $p = 0.0116$) and caspase 3 (by 83 and 90% in 1 and 3 h hypoxia, respectively; $F_{2,15} = 24.64$, $p < 0.0001$), and BNIP3 (Fig. 4, see above), while BAX and Bcl-2 remained unchanged. The apoptotic indicator AIF was also unchanged by hypoxia.

Finally, to examine whether hypoxia-induced degradation of thermogenic proteins observed in naked mole-rats occurs also in iBAT of other closely related mole-rat species, we also examined the effect of hypoxia on the expression of UCP1 and OXPHOS proteins in iBAT dissected from *Cryptomys hottentotus mahali* (CHM), *Cryptomys hottentotus pretoriae* (CHP), *Cryptomys hottentotus hottentotus* (CHH) and *Georychus capensis* (GC) following in vivo treatment in either normoxic or hypoxic conditions (Figs. 9 and 10;

found that, during hypoxia, mitochondrial area (as a % of the cytoplasm; $F_{3,110} = 19.01$, $p < 0.0001$), cristae length ($F_{3,104} = 23.53$, $p < 0.0001$), and cristae number ($F_{3,109} = 14.20$, $p < 0.0001$) were all decreased in hypoxia, whereas mitochondrial length was unchanged ($F_{3,134} = 0.77$, $p = 0.5155$). Notably, these ultrastructural changes were all partially or completely reversed within 1 h of recovery in normoxia.

Since hypoxia is a powerful regulator of ubiquitination and proteasomal degradation of proteins[28], we next examined the ubiquitination levels of iBAT proteins in naked mole-rats exposed to hypoxia in vivo. We found that the ubiquitination levels in hypoxic iBAT homogenates were significantly increased compared to homogenates from normoxic animals (Supplementary

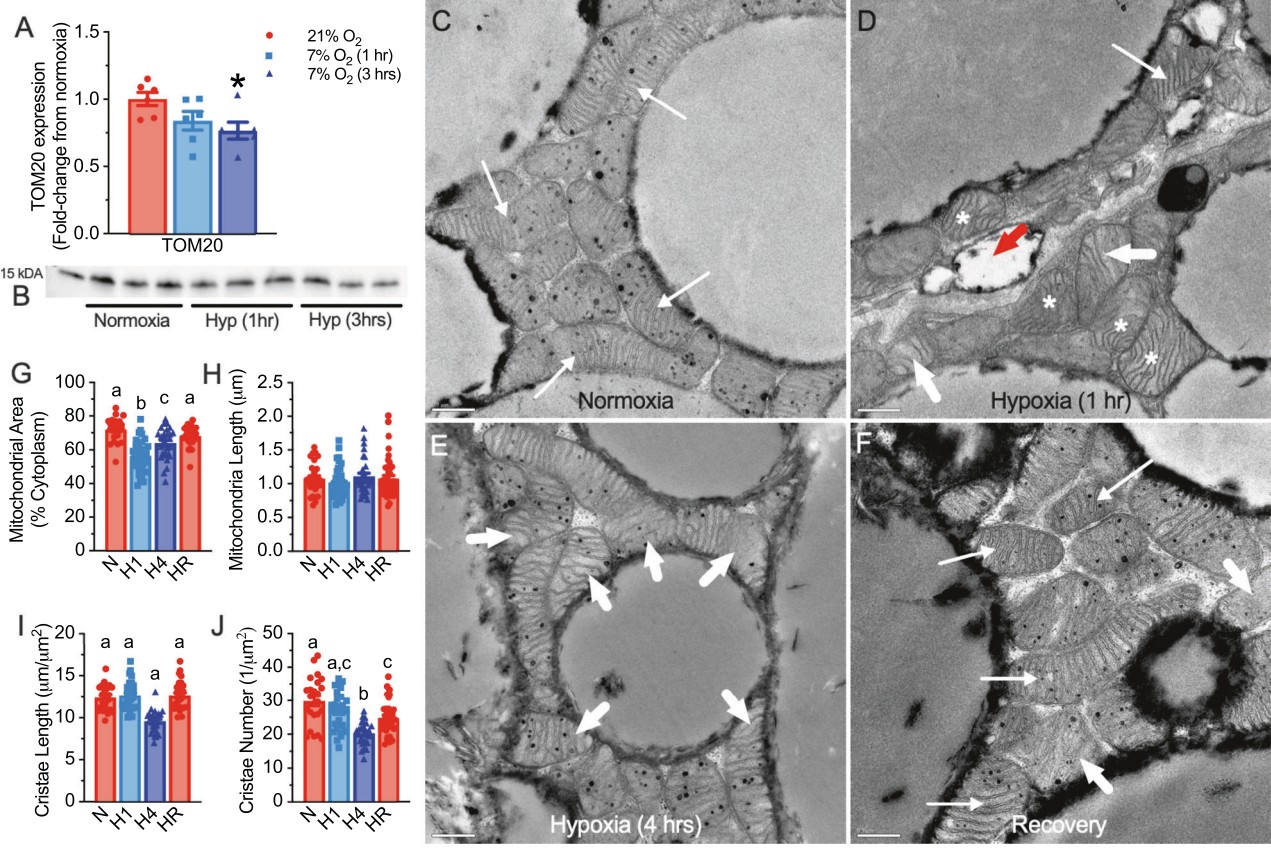

**Fig. 5 Mitochondrial and cristae density is reduced in acute hypoxia. A** Summary of translocase of outer membrane 20 (TOM20) protein expression in interscapular brown adipose tissue (iBAT) homogenates from naked mole-rats held at 30 °C in normoxia (21% $O_2$, red bars; $n = 6$ biologically independent samples) or after 1 or 3 h of hypoxia (7% $O_2$, light and dark blue bars, respectively; $n = 6$ biologically independent samples). **B** Western blot image of TOM20 protein bands. **C–F** Representative electron micrographs of iBAT from naked mole-rats held at 30 °C in normoxia (**C**), 1 h hypoxia (**D**), 4 h hypoxia (**E**), or 4 h of hypoxia with 1 h recovery in normoxia (**F**). Thin arrows indicate normal mitochondria, thick arrows indicate abnormal mitochondria, asterisks indicate tortuosity, and red arrowhead indicates degradative or lysosome-like vacuoles. Scale bars: 0.5 μm. **G–J** Summaries of mitochondrial area (**G**, $n = 29$, 36, 31, and 42 technical replicates for normoxia (N: red bars at left), 1 and 4 h of hypoxia (light and dark blue bars, and H1 and H4, respectively), and 4 h hypoxia with recovery (HR: red bars at right), respectively, from 4 independent animals for each condition) and average max length (**H**, $n = 22$, 33, 29, and 30 technical replicates for normoxia, 1 and 4 h of hypoxia, and 4 h hypoxia with recovery, respectively, from 4 independent animals for each condition), and cristae length (**I**, $n = 24$, 26, 24, and 29 technical replicates for normoxia, 1 and 4 h of hypoxia, and 4 h hypoxia with recovery, respectively, from 4 independent animals for each condition) and number (**J**, $n = 24$, 29, 26, and 34 technical replicates for normoxia, 1 and 4 h of hypoxia, and 4 h hypoxia with recovery, respectively, from 4 independent animals for each condition) from naked mole-rats treated as per (**C–F**). Data are mean ± SEM. Asterisks indicate significant difference from normoxic controls; letters indicate significant difference (one-way ANOVA with Dunnett's multiple comparisons test; $F_{2,15} = 3.98$, $p = 0.0428$ for TOM20; $F_{3,110} = 19.01$, $p < 0.0001$ for mitochondrial area, $F_{3,104} = 23.5_3$, $p < 0.0001$ for cristae length, and $F_{3,109} = 14.20$, $p < 0.0001$ for cristae number; $p = 0.5155$ for mitochondrial length). Source data are provided as a Source Data file.

$n = 4–6$ independent biological replicates per species). As in naked mole-rats, iBAT homogenates from CHH, CHM, and CHP also exhibited 32–65% decreases in UCP1 expression in hypoxia (CHH: $t(4) = 2.818$, $p = 0.0228$, CHM: $t(5) = 2.361$, $p = 0.0321$, CHP: $t(6) = 2.181$, $p = 0.0369$; Fig. 9). Conversely, UCP1 increased by 53% in GC iBAT, but this relationship did not reach statistical significance ($t(4) = 1.858$, $p = 0.0674$). On the other hand, the expression of OXPHOS proteins was generally unchanged by hypoxia, except for complexes III and IV in CHH, which were significantly increased by acute hypoxia ($t(6) = 2.516$, $p = 0.0445$ for CIII, $t(4) = 2.876$, $p = 0.041$; Fig. 10).

## Discussion

In the present study, we demonstrate that acute hypoxia diminishes interscapular thermogenesis in naked mole-rats, consistent with previous observations of substantial hypoxia-mediated suppression of whole-body metabolic rate and $T_b$ in this species[13,14], and describe a mechanism by which thermogenic and metabolic

proteins in iBAT may be downregulated during hypoxia. Specifically, our study provides evidence that in naked mole-rats, hypoxia: (1) decreases interscapular thermogenesis at temperatures near or below thermoneutrality, and (2) causes a significant reduction in iBAT levels of UCP1 and OXPHOS proteins, which may be (3) mediated by a canonical mitochondrial fission pathway, the activation of which is accompanied by changes in mitochondrial content and cristae ultrastructure. Taken together, our results demonstrate that hypoxia depresses iBAT thermogenesis in naked mole-rats via unique mechanisms that are mediated, at least in part, by targeted reductions in mitochondrial content and of the protein that is essential for NST, UCP1. Furthermore, acute hypoxia (4) decreases UCP1, but not OXPHOS protein expression in iBAT from three social African mole-rat species but not a solitary species, suggesting that this response may be conserved in social members of this rodent lineage.

Hypoxia reduces thermogenesis in other mammalian models of hypoxia-tolerance, including hibernators and neonates[30,31], and

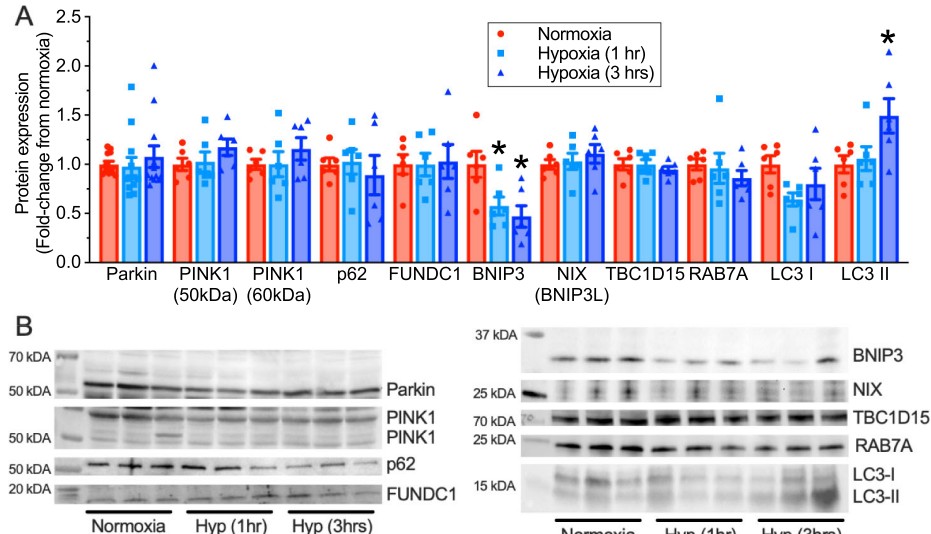

**Fig. 6 Hypoxia does not alter the expression of key mediators of ubiquitin- or receptor-mediated mitophagy. A** Summary of changes in receptor-mediated mitophagy-related protein expression in interscapular BAT (iBAT) homogenates from naked mole-rats held at 30 °C in normoxia (21% $O_2$, red bars) or after 1 or 3 h of hypoxia (7% $O_2$, light and dark blue bars, respectively; $n = 12$ biologically independent samples for Parkin and 6 biologically independent samples for all other proteins). **B** Western blot images of protein bands from (**A**). Data are mean ± SEM. Asterisks indicate significant difference from normoxic controls (one-way ANOVA with Tukey multiple comparisons test; $F_{2,15} = 6.16$, $p = 0.0111$ for Bcl-2 adenovirus E1B 19 kDa-interacting protein 3 (BNIP3); $F_{2,12} = 4.14$, $p = 0.0441$ for light chain 3 II (LC3II); $p > 0.05$ for FUN14 domain containing 1 (FUNDC1), sequestosome 1 (p62), phosphatase and tensin homolog (PTEN)-induced kinase 1 (PINK1), ras-related protein 7A (RAB7A), and TBC1 domain family member 15 (TBC1D15)). Additional abbreviations: N – normoxia, H1 – 1 h hypoxia, H3 – 3 h hypoxia. Source data are provided as a Source Data file.

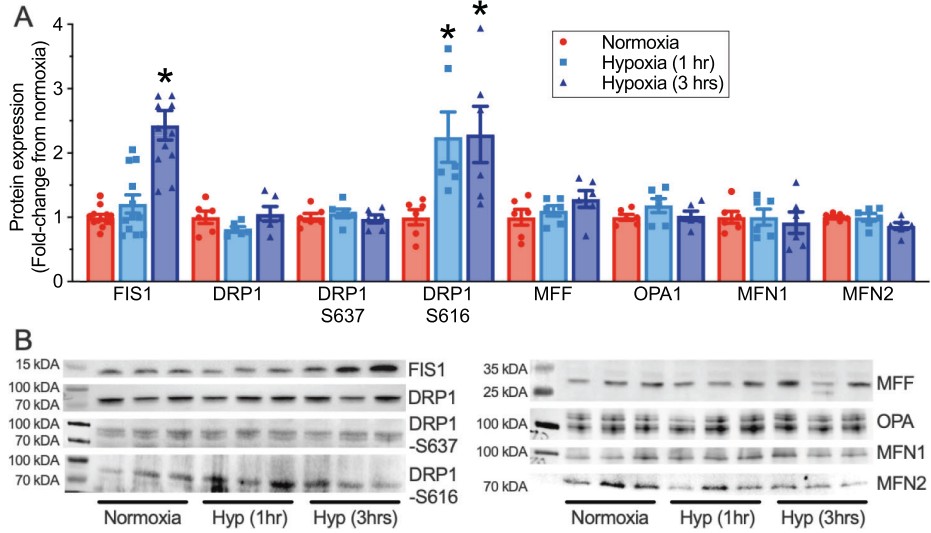

**Fig. 7 Hypoxia activates mitochondrial fission, but not mitochondrial fusion. A** Summary of the expression of mitochondrial fission proteins in interscapular brown adipose tissue (iBAT) homogenates from naked mole-rats held at 30 °C in normoxia (21% $O_2$, red bars) or after 1 or 3 h of hypoxia (7% $O_2$, light and dark blue bars, respectively; $n = 12$ biologically independent samples for FIS1 and 6 biologically independent samples for all other proteins). **B** Western blot images of protein bands from (**A**). Data are mean ± SEM. Asterisks indicate significant difference from normoxic controls (one-way ANOVA with Tukey multiple comparisons test; $F_{2,33} = 23.83$, $p < 0.0001$ for fission protein 1 (FIS1); $F_{2,15} = 4.49$, $p = 0.0296$ for dynamin-related protein 1-S616 (DRP1-S616); $p > 0.05$ for mitochondrial fission factor (MFF), mitofusin 1 and 2 (MFN1 or 2), and optic atrophy 1 (OPA1)). Source data are provided as a Source Data file.

in some hypoxia-intolerant adult rodents; however, the mechanisms underlying these decreases are varied and do not involve changes in NST, or if they do, are not mediated by changes in thermogenic protein expression. For example, many mammals utilize a variety of behavioral and physiological strategies (e.g., reduced huddling, moving to a cooler environment, etc.) to lower $T_b$ and reduce thermogenic energy consumption, and thus conserve oxygen, in hypoxia[32–34], while others decrease shivering thermogenesis in hypoxia[35–37]. Conversely, while NST is also reduced in hypoxia in other small mammals[38,39], this response seems to be largely dependent on a reset of the hypothalamic thermogenic threshold and thus of the neural activation of iBAT activity[25,40,41], rather than a downregulation of thermogenic effectors.

In most small mammals, BAT thermogenesis is an energy-intensive process that can drastically impact whole-body energy

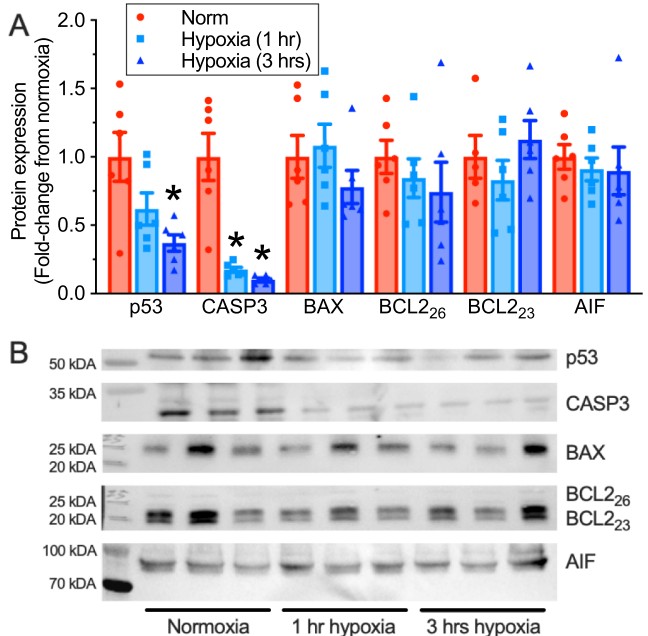

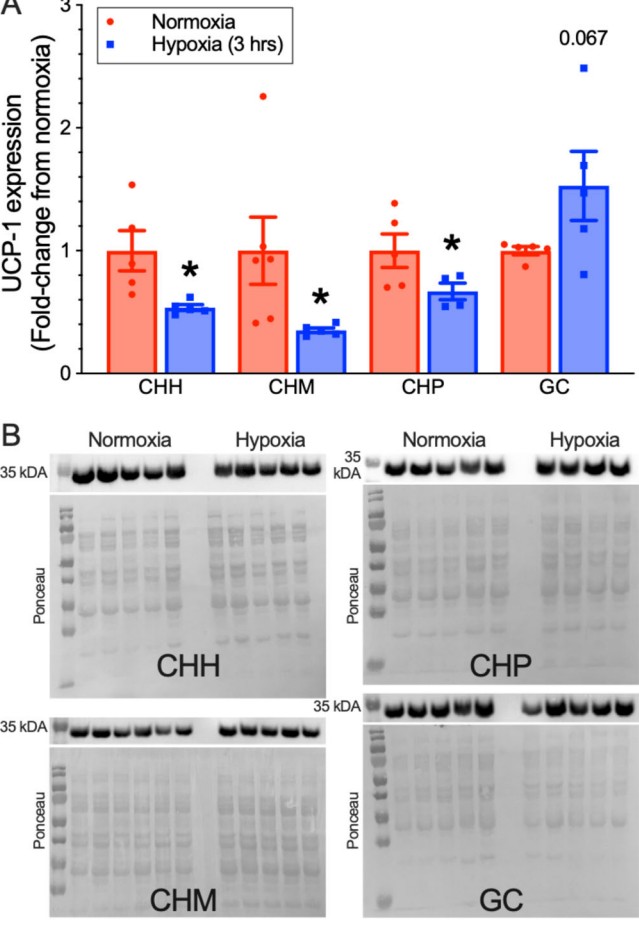

**Fig. 8 Apoptosis is not activated during hypoxia. A** Summary of the expression of apoptosis regulatory proteins in interscapular brown adipose tissue (iBAT) homogenates from naked mole-rats held at 30 °C in normoxia (21% $O_2$, red bars; $n = 6$ biologically independent samples) or after 1 or 3 h of hypoxia (7% $O_2$, light and dark blue bars, respectively; $n = 6$ biologically independent samples). **B** Western blot images of protein bands from (**A**). Data are mean ± SEM. Asterisks indicate significant difference from normoxic controls (one-way ANOVA with Tukey multiple comparisons test; $F_{2,15} = 6.087$, $p = 0.0116$ for p53; $F_{2,15} = 24.64$, $p < 0.0001$ for caspase 3 (Casp3); $p > 0.05$ apoptosis inducing factor (AIF), Bcl-2-associated X (BAX), and B-cell lymphoma 2 (Bcl-2)). Source data are provided as a Source Data file.

**Fig. 9 Acute hypoxia reduces uncoupling protein 1 (UCP1) expression in interscapular brown adipose tissue (iBAT) from social, but not solitary species of African mole-rats. A** Summary of western blot analysis of UCP1 protein expression in interscapular BAT homogenates from 4 species of African mole-rats related to naked mole-rats (CHH: *Cryptomys hottentotus hottentotus*, CHM: *Cryptomys hottentotus mahali*, CHP: *Cryptomys hottentotus pretoriae*, GC: *Georychus capensis*) treated in 28 °C in normoxia (18% $O_2$, red bars; $n = 6$ biologically independent samples for CHM and 5 for other species) or after 3 h of hypoxia (5% $O_2$, blue bars; $n = 4$ biologically independent samples for CHP and 5 for other species). Data are mean ± SEM. Asterisks indicate significant difference from normoxic controls (Welch's one-sided *t*-test; CHH: $t(4) = 2.818$, $p = 0.0228$, CHM: $t(5) = 2.361$, $p = 0.0321$, CHP: $t(6) = 2.181$, $p = 0.0369$, GC: $t(4) = 1.858$, $p = 0.0674$). **B** Western blot images of UCP1 protein expression from (**A**), and total protein gel expression quantified with Ponceau S. Source data are provided as a Source Data file.

homeostasis[25,41]. Previous studies indicate the existence of functional iBAT in naked mole-rats[16,18,42]. Furthermore, activation of BAT in naked mole-rats greatly increases whole-animal oxygen consumption, and this increase explains most of the increases in whole-body oxygen consumption in naked mole-rats under these conditions[42–44]. Our study confirms this knowledge: we show that naked mole-rats actively thermoregulate and engage BAT thermogenesis at temperatures near or below thermoneutrality or following treatment with an adrenergic agonist.

Beyond normoxic conditions, our work demonstrates that hypoxia rapidly decreases BAT thermogenesis in naked mole-rats. Notably, we observe a hypoxia-induced decrease in BAT thermogenesis in animals that have received injections of either saline or isoproterenol, and in environmental temperatures near or well below the thermoneutral zone of this species. These results indicate that hypoxia diminishes BAT thermogenesis in naked mole-rats regardless of the degree to which BAT is activated. But, how are these changes regulated?

UCP1 is essential for BAT thermogenesis[45–47], is expressed in mitochondria from classical BAT and is located in the mitochondrial inner membrane[23,24]. It is estimated that UCP1 comprises approximately 10% of BAT mitochondrial protein content[27], and it is well established that defects in the activity of UCP1 resulting from post-translation modifications or the absence of UCP1 can impact BAT thermogenesis[46,48,49]; mice lacking UCP1 are cold intolerant[46]. Given this essential role for UCP1 in BAT thermogenesis, we examined the effect of acute hypoxia on UCP1 protein levels in iBAT and found significant decreases following only 1 h of in vivo hypoxia, and that

these decreases are sustained after 3 h. This change is likely primarily due to reduced, but not eliminated expression of UCP1 in iBAT cells as immunohistochemistry analysis reveals that the number of UCP1-positive cells decreases only mildly relative to the more robust change in a quantitative measure of total iBAT UCP1 expression (i.e., western blot data). Conversely, UCP3, which also plays a role in thermogenesis[50], did not change significantly with progressive hypoxia.

In other rodent species, changes in UCP1 expression require longer-term hypoxic exposure to manifest. For example, exposure of cultured and immortalized mouse SV40T fetal brown inguinal adipocytes to short-term hypoxia (1, 12, or 24 h in 1% $O_2$) has no impact on UCP1 protein expression[51]. Conversely, longer-term hypoxia alters UCP1 function in some mammals. For example, (i)

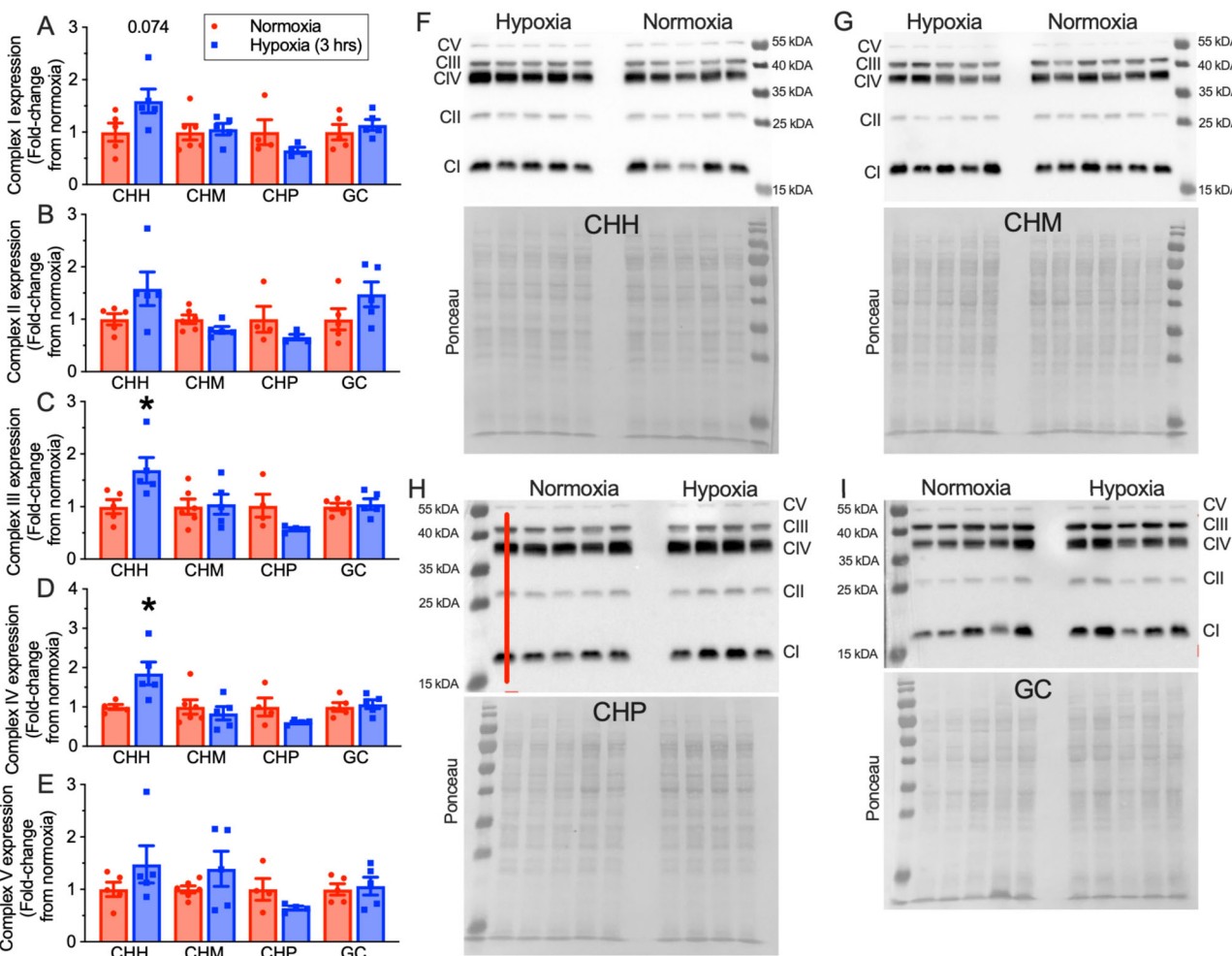

**Fig. 10 Hypoxia does not consistently modify electron transport chain (ETC) protein expression in cousin species of African mole-rat.** A–E Summaries of western blot analysis of ETC (CI–CIV) and $F_1F_o$-ATPase (CV) protein expression in interscapular brown adipose tissue (iBAT) homogenates from 4 species of African mole-rats related to naked mole-rats (CHH: *Cryptomys hottentotus hottentotus*, CHM: *Cryptomys hottentotus mahali*, CHP: *Cryptomys hottentotus pretoriae*, GC: *Georychus capensis*) treated in 28 °C in normoxia (18% $O_2$, red bars; $n = 6$ biologically independent samples for CHM, 5 for GC and CHH, and 4 for CHP) or after 3 h of hypoxia (5% $O_2$, blue bars; $n = 4$ biologically independent samples for CHP and 5 for other species). Data are mean ± SEM. Asterisks indicate significant difference from normoxic controls (Welch's one-sided *t*-test; $t(6) = 2.516$, $p = 0.0445$ for CHH CIII and $t(4) = 2.876$, $p = 0.041$ for CHH CIV; $p > 0.05$ for all other species and complexes). F–I Western blot images of ETC and $F_1F_o$-ATPase protein expression, and total protein gel expression quantified with Ponceau S, from CHH, CHM, CHP, and GC. Note that lane 1 of the CHP gel erroneously contains a mouse sample and was excluded from analysis. Source data are provided as a Source Data file.

4–6 days of chronic hypoxia results in significant decreases in UCP1 protein expression in neonatal and adult BAT in rats[52,53], (ii) 4 weeks of chronic hypoxia increases UCP1 mRNA, but not UCP1 protein expression in mice[54], and (iii) in a mouse model of sleep apnea, 37 days of daily 8 h periods of intermittent hypoxia decreases UCP1 mRNA expression and UCP1-positive cells in iBAT[55]. While this sparse experimental evidence indicates that hypoxia can downregulate UCP1 expression in hypoxia-intolerant species, the time required for this process ranges from several days to several weeks. This is likely because a relatively long time is required to turn over UCP1 in most mammals, the rate of which depends on the metabolic status of BAT: in mouse brown adipocytes UCP1 turnover takes 3.7 ± 0.4 days in basal conditions and 8.4 ± 0.9 days with chronic adrenergic stimulation[56], with a half-life of ~ 30 h in basal conditions[57,58]. Thus, the downregulation in UCP1 expression in naked mole-rat iBAT after 1 h of hypoxia is remarkably rapid compared to other small rodent species.

The rapid degradation of thermogenic proteins in iBAT of naked mole-rats could be mediated by several well-known and -characterized mechanisms, including the ubiquitin–proteasome system (UPS), mitophagy, or mitochondrial fission/fusion events. In UPS-dependent pathways, targeted proteins are ligated with ubiquitin and then degraded by proteasomes. Importantly, protein ubiquitination is a post-translational modification that leads to the degradation of protein not only through proteolysis but also through mitophagy. These mechanisms are activated by hypoxia in some models[59], and naked mole-rats have a very high rate of autophagy compared to other rodents[60]. However, although we observe a global increase in ubiquitination in naked mole-rat iBAT after 1 h of hypoxia and a general increase in LC3II activation at 3 h of hypoxia, we did not detect co-immunoprecipitation of ubiquitin with UCP1, nor changes in any UPS- or mitophagy-related proteins (Parkin, PINK1, p62, FUNDC1, BNIP3, Nix, or TBCD15). Taken together, these results suggest that the UPS or mitophagy mechanisms are not mediating the downregulation of UCP1 in iBAT.

In addition to UCP1, hypoxia also decreases the levels of mitochondrial OXPHOS proteins in naked mole-rat iBAT. The rate of turnover of proteins in these complexes in mice can vary

depending on the tissue or treatments that cause metabolic changes in mitochondria[61], but are in general considerably slower than the rate that we observe in naked mole-rat iBAT. Given the rapid decreases in UCP1 and these mitochondrial metabolic proteins in iBAT during hypoxia, and the absence of ubiquitin-binding to naked mole-rat iBAT UCP1, we considered changes in mitochondrial structure and ultrastructure, which can be potent regulators of thermogenic potential[62].

Fission is a complex process in mitochondrial regulation and plays key roles in mitochondrial division and in the initiation and spatial organization of mitophagy[63]. Mitochondrial fission is mediated by Drp1, with phosphorylation at site S616 associated with translocation of Drp1 to the mitochondrial membrane and activation of fission, and dephosphorylation at site 637 associated with deactivation of Drp1[64–66]. Despite significant research attention, the mechanism of mitochondrial fission remains poorly understood. Using super-resolution microscopy, a recent study identified two spatially distinct modes of mitochondrial fission, both of which are mediated by Drp1[67]. Midzone division is associated with fission and reorganization of healthy mitochondria, whereas peripheral division is associated with the budding-off of damaged portions of mitochondria. Importantly, both forms of fission are associated with Drp1 accumulation, but are regulated by association with different cellular factors. Specifically, fission of healthy mitochondria is regulated by contact with the ER, whereas fission of damaged mitochondria is associated with Fis1 upregulation. Downstream of Drp1/Fis1-mediated fission, damaged mitochondrial segments undergo mitophagy, mediated by interactions with Parkin/PINK1[68–70], and thus whole segments of mitochondria become ubiquitylated and tagged for lysosomal degradation.

Our observations that Drp1 is phosphorylated at site S616 but not S637 at both 1 and 3 h of hypoxia, and that mitochondria exhibit rapid and reversible changes in ultrastructure and cristae morphology on the same timeline, indicate that fission is activated by hypoxia in naked mole-rat iBAT. Furthermore, the absence of any change in Fis1 expression at 1 h suggests that fission of healthy mitochondria occurs rapidly during acute hypoxia[67]. The mitochondrial inner membrane is folded to form cristae, and this is where UCP1 and OXPHOS proteins are located. Therefore, fission of healthy mitochondria may mediate the decrease in thermogenic proteins from the inner mitochondrial membrane and facilitate the rapid decrease in iBAT thermogenesis during acute hypoxia. Notably, we observe a marked increase in Fis1 following 3 h of acute hypoxia, suggesting that mitochondrial degradation occurs with longer-term hypoxia and that peripheral fission of damaged portions of mitochondria occurs with prolonged hypoxia. Presumably with longer-term experiments, we would observe an upregulation in the Parkin/PINK1 mitophagy pathway, and indeed we see a weak, but insignificant trend towards such activation in the expression of both Parkin and the two components of PINK1, along with a significant increase in LC3-II at 3 h of hypoxia. However, in their natural environments, naked mole-rats typically do not spend more than a few hours in the nest chamber at a time before moving about the colony to perform various tasks, and burrows are not expected to be hypoxic outside of these crowded nest chambers[71], thus animals would likely not experience such prolonged durations of acute and severe hypoxia in nature.

Naked mole-rats are members of a large family of primary subterranean rodents, many of which live in variably social groupings that might be expected to generate similar patterns of severe but intermittent hypoxia in their natural habitat. Therefore, we expanded our investigation to examine the turnover of UCP1 and ETC proteins after hypoxia exposure in four other hypoxia-tolerant African mole-rat species related to naked mole-rats, including three social species (CHH, CHM, CHP) and a solitary species (GC). Intriguingly, hypoxia causes distinct patterns of change in the expression of thermogenic proteins in these four mole-rat species that are partially divergent from the pattern observed in naked mole-rats. Specifically, hypoxia decreases the expression of UCP1 in the social, but not the solitary mole-rat species, whereas hypoxia did not consistently modify ETC proteins in any species.

These differences may be due to the social structure within which these species commonly live. For example, cooperatively breeding and eusocial African mole-rat species, which include naked and Damaraland mole-rats (*Fukomys damarensis*), have relatively large colony sizes[72] and construct complex tunnel systems with a relatively warm but nonetheless variable temperature range (19.6–29.3 °C for Damaralands[73]). The number of colony members within a burrow system likely plays a critical role in the respiratory gas concentrations and the temperatures individual mole-rats experience. For example, $CO_2$ concentrations in the burrow system of solitary GC mole-rats are ~1.2%, whereas eusocial Damaraland mole-rats, which live in colonies of up to 40 individuals, have $CO_2$ concentrations as high as 6.4%[73]. Furthermore, even though the environmental temperatures experienced are similar between the solitary GC and eusocial Damaraland mole-rats (and also naked mole-rats), individual Damaraland and naked mole-rats likely experience a smaller true temperature range as they employ huddling behaviors in the nest chamber (a portion of the burrow system which is expected to have the lowest $O_2$ concentration, especially during huddling)[74]. Along this gradient of social organization lie several social species, such as the CHH, CHP, and CHM mole-rats investigated in this study. The maximum colony size for these species is ~20 individuals with a mean colony size of 7[75,76]. Thus, the respiratory gas concentrations of their burrow systems are expected to fall in between the solitary and eusocial species; however, this has not been documented to date. Therefore, the variation seen between naked mole-rats, CHH, CHP and CHM, and GC, may reflect an inverse relationship between the severity of hypoxia and the thermal range experienced by an individual mole-rat, which is determined by the number of colony members or an individual within a burrow system. Taken together, these results suggest that the rapid cycling and downregulation of thermogenic proteins in naked mole-rat iBAT is a mechanism that is shared between social mole-rat species, whereas the rapid alterations in ETC proteins is not.

Overall, our results support the general conclusion that naked mole-rats and other social mole-rat species respond to acute hypoxia by diminishing iBAT thermogenesis and that, in naked mole-rats, this is partially achieved by acutely decreasing the levels of thermogenic proteins and altering mitochondrial ultrastructure. In addition, we provide a plausible mechanism via which fission of healthy mitochondria may mediate this process. Importantly, the absence of mitophagy and apoptosis, and the reversibility of the suppression of iBAT thermogenesis and mitochondrial morphological changes following reoxygenation, indicate that this process occurs in healthy mitochondria and is not deleterious to mitochondrial function. Further elucidation of adaptive mechanisms in iBAT of naked mole-rats will lead to a better understanding of potentially unique mechanisms within this tissue that allow these interesting animals to survive in extreme conditions. It is likely that decreases in iBAT activity (and associated thermogenesis) occur as part of a larger suite of adaptive mechanisms during hypoxia, along with many other means of energy (and thus $O_2$) conservation; however, given the high metabolic cost of thermoregulation in small rodents, the mechanism we identify is likely of significant importance to the hypoxia-tolerance of naked mole-rats.

## Methods

**Animals**. Naked mole-rats were bred at the University of Ottawa and group-housed in interconnected multi-cage systems at 30 °C and 21% $O_2$ in 50% humidity with a 12 L:12D light cycle. CHM, CHP, GC, and CHH were wild captured in South Africa and were individually housed at the University of Pretoria at ~26 °C and 18% $O_2$ in 50% humidity with a 12 L:12D light cycle. Animals were fed fresh tubers, vegetables, and fruit, and Pronutro cereal supplement (Bokomo Food Products, Namibia) ad libitum. Animals were not fasted prior to experimental trials.

All experimental procedures were approved by the University of Ottawa Animal Care Committee (protocol #2535), in accordance with the Animals for Research Act and by the Canadian Council on Animal Care. Trapping and experiments conducted in South Africa were conducted under appropriate permits issued by Cape Nature Conservation and the Department of Nature Conservation in the Western Cape, Republic of South Africa (CN44-31-2285) and with experimental procedures approved by the animal ethics committee of the University of Pretoria (EC069-17). All experiments were performed during daylight working hours in the middle of the animals' 12 L:12D light cycle. Naked mole-rats that are housed within colony systems do not exhibit circadian rhythmicity of general locomotor activity[77], and exhibit inconsistent rhythmicity of $T_b$ and metabolic rate[78]; however, significant changes in these latter parameters were only reported in animals during the nocturnal phase of their circadian cycle with no significant changes observed during the daylight period of this cycle. Therefore, since we only ran experimental trials during the daylight period, we do not expect our results to be influenced by circadian rhythms. We did not conduct in vivo experiments in the other mole-rat species.

We examined physiological responses to environmental hypoxia and/or collected tissue from 135 male and female non-breeding subordinate adult (1–2-year-old) naked mole-rats weighing 48.3 ± 4.9 g (mean ± SEM). Non-breeding (subordinate) naked mole-rats do not undergo sexual development or express sexual hormones and thus we did not take sex into consideration when evaluating our results[79]. We also collected tissues from 8–10 animals per species of CHM (109.1 ± 6.6 g), CHP (114.8 ± 12.3 g), GC (133.2 ± 19.0 g), and CHH (78.7 ± 5.5 g), following 3 h of normoxia or hypoxia (see below). The age and sex of these wild-caught animals was not determined. All animals underwent a single experimental protocol (see below), to avoid artefacts due to acclimation to handling or hypoxia.

**Experimental design and tissue collection**. Surface temperatures were measured via thermal imaging from 47 naked mole-rats divided into five experimental groups: 20 °C, 30 °C, 30 °C + sham saline injection, 30 °C + isoproterenol injection (2.5 mg/kg), and 36 °C ($n = 8$, 11, 12, 5, and 11 individual animals, respectively). For all groups (and following saline or saline + drug injections where applicable), baseline recordings were obtained for 1 h in normoxia (21% $O_2$, 0% $CO_2$, balance $N_2$) and then the incurrent gas composition was switched to 7% $O_2$ (0% $CO_2$, balance $N_2$) for 1 h followed by 1 h in normoxia (recovery). Following experimentation, animals were returned to their colonies and were not used again for this study. The temperature of the room in which the experiments were conducted was held at 20, 30, or 36 °C, as appropriate, and animals were acclimated for at least 2–3 h at the specified temperature prior to commencing experimentation. These temperatures were selected since an $T_a$ of 30 °C is the housing temperature of our colonies, and is near the thermoneutral zone of naked mole-rats (which spans from ~30.5–34 °C)[80]. The 20 °C experimental temperature was selected to increase the thermal scope within which the animals were able to respond through thermo-regulatory adaptations to hypoxia. Naked mole-rats have a higher metabolic rate in colder temperatures relative to near their thermoneutral zone[14], and thus repeating our experiments in this temperature magnifies the impact of our treatments on metabolic rate and $T_b$ and therefore our ability to detect any physiological changes in this condition. The 36 °C experimental temperature was selected to examine thermogenic responses in an $T_a$ at which naked mole-rats would likely not need to actively thermoregulate (i.e., above their thermoneutral zone).

In other experiments, 74 naked mole-rats were exposed to either normoxia or 1 or 3 h of acute hypoxia (7% $O_2$). All of these experiments were conducted at an $T_a$ of 30 °C. Similarly, captured populations of CHM, CHP, GC, and CHH were divided into two treatment groups ($n = 4$–5 per treatment per species) and exposed to 3 h of normoxia or hypoxia (5% $O_2$) at ~28 °C. Note that these experiments were conducted in Pretoria, which is at mild altitude, resulting in normoxic ambient oxygen level of 18 kPa (~18% $O_2$, whereas the partial pressure of oxygen in Ottawa is 20.79% $O_2$). At the end of each exposure, animals were sacrificed as described below, with tissue collected over ice and stored at −80 °C until analyzed.

**Collection and analysis of thermographic data**. FLIR thermal images were captured directly to radiometric video files using an infrared thermal imaging camera (Make: FLIR, Model: SC 660, Teledyne FLIR, LLC, Wilsonville, OR, USA) connected to a computerized acquisition program (Thermacam Researcher Pro v 2.9, Teledyne FLIR). Images were captured every 10 s throughout the experimental period. Image analysis was as described previously[81–83]. Briefly, emissivity was assumed to be 0.96, air temperature and reflected environment temperatures were set to 20, 30, or 36 °C, as appropriate, relative humidity set to 50%, the object distance set to 0.35 m, and the transmittance of the Germanium IR window set to

0.95 (determined empirically). Image analysis was conducted by a blinded researcher using images taken every fifth minute during the experimental period, and by drawing regions of interest (ROI) over the interscapular and hind back regions. Average temperatures across the entire ROI were extracted and quantified. Data were corrected using the Thermimage package V3.0 in R.

Body temperature measurements. Body temperature was measured using a handheld RFID reader that scanned individual naked mole-rats instrumented with subcutaneous RFID microchips (Destron Fearing, Dallas, TX) every 10 min, with microchip implantation and validation described previously[14].

**Western blotting**. Interscapular BAT was dissected from mole-rats that were kept either in normoxia (21% $O_2$) or in hypoxia (5% or 7% $O_2$) for 1 or 3 h, as indicated in the "Results" section. After dissection, iBAT was snap-frozen and stored at −80 °C until it was used for the protein expression analysis. On the day of the experiment, tissues from 4–12 different animals (as indicated in the "Results" section) for each treatment condition were randomly selected from our tissue pool and the frozen tissue was thawed on ice, cleaned from white fat, cut into small pieces, and subsequently homogenized at 100 mg/ml (W/V) in Radio-Immunoprecipitation Assay (RIPA) buffer with protease inhibitor mixture (ThermoFisher Scientific, Ottawa, ON, Canada) and phosphatase inhibitors (BOSTER). iBAT was manually homogenized on ice in a Potter Elvehjem Teflon–glass tissue grinder using 4–5 strokes. The homogenate was agitated at 100 rpm on ice for 1 h and then spun at 12,000$g$ for 10 min at 4 °C to remove the solidified lipid layer. The remaining supernatant was kept for protein concentration measurements using the Biuret assay. First, 10–50 μg of protein was loaded per lane onto a 12–15% SDS-polyacrylamide gel, which in turn was run for 1–1.5 h at 130 V. The separated proteins were then transferred to a nitrocellulose membrane for 1 h at 100 V. Successful transfer was determined by Ponceau S staining. Membranes were then incubated with 5% bovine serum albumin for 30 min to block non-specific binding sites. Levels of specific proteins were determined using the following primary antibodies: UCP1 antibody (1:3000; Sigma Aldrich, #U6382), total OXPHOS rodent cocktail (1:2000; Abcam, #Ab110413), anti-ubiquitin (1:2000; Abcam, #Ab7780), LC3B (1:1000; Cell Signaling Technology, #2775), PARKIN (1:2000; ABclonal, #A0968), p62 (1:1000; ABclonal, #A11483), PINK1 (1:2000; ABclonal, #A7131), FUNDC1 (1:2000; ABclonal, #16318), BNIP3 (1:1000; Abcam, #Ab10433), BNIP3L (NIX; 1:250; Santa Cruz, #SC-166314), UCP3 (1:1000; Abcam, #AB3477), Fis1 (1:1000; BioVision, #3491), Drp1 (1:1000; BD Biosciences, #611118), Drp1-S616 (1:1000; ABclonal, #AP0849), Drp1-S637 (1:1000; Cell Signaling Technology, #4867), Casp3 (1:1000; ABclonal, #A0214), p53 (1:200; Cell Signaling Technology, #2524), AIF (1:2000; Abcam, #ab32516), BAX (1:500; Santa Cruz, #SC-526), Bcl2 (1:1000; Santa Cruz, #sc-492), Mff (1:1000; ABclonal, #A4874), Opa1 (1:2000; ABclonal, #A9833), Mfn1 (1:2000; ABclonal, #A9880), Mfn2 (1:1000; ABclonal, #A19678), TOM20 (1:2000; ABclonal, #A19403), TBC1D15 (1:2000; ABclonal, #A10593), RAB7A (1:2000; ABclonal, #A12784), and anti-ubiquitin antibody (1:2000; Abcam, #Ab7780). Quantification was performed using ImageJ software (V1.53j, NIH, Bethesda, MD).

**TEM**. Sixteen naked mole-rats were randomly divided into four treatment groups ($n = 4$ each), including: control (normoxia), 1 or 4 h of 7% hypoxia, or 4 h of 7% hypoxia followed by 1 h of recovery in normoxia. Immediately following treatment, animals were anesthetized with ketamine (200 mg/kg) + xylazine (10 mg/kg) and then rapidly returned to their treatment condition until they had lost consciousness, to minimize reoxygenation of hypoxic animals. Animals were then perfusion-fixed via cardiac puncture using 2% formaldehyde and 2.5% glutaraldehyde. Following fixation, iBAT was dissected and further fixed overnight at 4 °C in 2.5% glutaraldehyde in 0.15 M sodium cacodylate buffer, pH 7.4, and washed three times with washing buffer. Samples were post-fixed with 1% aqueous $OsO_4$ + 1.5% aqueous potassium ferrocyanide for 2 h and washed three times with washing buffer. Specimens were dehydrated in a graded ethanol-$dH_2O$ from 30%, 50%, 70%, 80%, 90%, to 100% ethanol. The samples were infiltrated with a graded Epon-ethanol series (1:1, 3:1) and then embedded in 100% Epon and then polymerized in an oven at 60 °C for 48 h. Ultra-thin sections (90–100 nm thick) were prepared from the polymerized blocks with a Diatome diamond knife using a Leica Microsystems EM UC7 ultramicrotome (Leica Biosystems, Buffalo Grove, IL, USA), transferred onto 200-mesh copper grids, and stained with 4% uranyl acetate for 6 min and Reynold's lead for 5 min. The TEM grids were imaged by a FEI Tecnai G[2] Spirit 120 kV TEM equipped with a Gatan Ultrascan 4000 CCD Camera Model 895. The proprietary Digital Micrograph 16-bit images (DM3) were converted to unsigned 8-bit TIFF images. Image analysis was performed blinded and using ImageJ (NIH) on micrographs obtained at 9000X (9k) or 19,000X (19k) magnifications. Metrics quantified included (i) total mitochondrial area normalized to cytoplasm area (%) from 9k micrographs, (ii) maximal length (mm) of individual mitochondria average from 19k micrograph, as well as (iii) average cristae number (1/mm$^2$) and total length (mm/mm$^2$) normalized to mitochondrial area from 19k micrographs. All metrics were quantified manually from scaled micrographs, using tools of Polygon and Freehand Selections (mitochondrial and cytoplasmic area; cristae length) or Straight Line (mitochondrial max length), followed by the Measure function.

**Immunohistochemistry**. Interscapular BAT was collected from naked mole-rats treated in normoxia ($n = 12$), or 1 or 3 h of hypoxia ($n = 11$ each) and samples prepared as described previously[49]. Briefly, iBAT was dissected, cleaned to remove any white adipose, muscle or connective tissues, and then fixed in 10% formalin overnight. iBAT was then ethanol dehydrated and stored in 70% ethanol prior to paraffin embedding. Embedded tissue was sectioned to the largest surface area. IHC staining was performed on 5 μm sections using the Leica Bond™ system using a modification of protocol F that eliminates the post primary step when using rabbit antibodies on rodent tissue. Sections were pretreated using sodium citrate buffer (pH 6.0, epitope retrieval solution 1) for 20 min. The sections were then incubated using a 1:1000 dilution of rabbit UCP-1 for 30 min at room temperature and detected using an HRP conjugated compact polymer system. Slides were then stained using DAB as the chromogen, counterstained with hematoxylin, mounted and cover slipped.

After staining, Mirax Viewer Image software (version 1.6) was used to analyze the sections in a ZEISS-MIRAX Midi Slide scanning system (Zeiss Microimaging, Oberkochen, Germany, and 3DTech, Budapest, Hungary) and digital images were acquired at 20x magnification. Images were extracted using Aperio ImageScope software (version 12.3.3; Leica Biosystems), and were then processed using Zeiss software ZEN 3.2 (Zen Lite; Carl Zeiss Canada Ltd, Toronto, Canada).

For LD analysis of hematoxylin and eosin (H&E)-stained sections, ROIs were carefully selected in each image using the rectangular selective tool bar in the ZEN 3.2 software, making sure to avoid blood vessels and edges of the tissue. All the ROIs were converted to a TIFF format. Then, images were analyzed using FIJI (ImageJ version 1.53j; NIH) by a blinded researcher. The range thresholding applied across all replicate images in all conditions was 200–255 and the range of LD area analyzed was between 1 and 1000 μm². The scale of each image processed was calibrated by adding a scale according to the ROI analyzed. Finally, the total LD area of each image was measured, and a percent lipid area was calculated using FIJI[84].

For UCP-1 cell count analysis, DAB-stained images were converted into TIFF format and then analyzed using FIJI by a blinded reviewer. For each replicate, two sections of 2000 × 2000 pixels were randomly taken and analyzed by manual counting of DAB-positive cells. This count was then divided by the total number of cells in each section to obtain the percentage of UCP-1-positive cells.

**Data collection and statistical analysis**. One-factor ANOVAs were used to examine the main effects of acute inspired $PO_2$ on surface temperature and $T_b$ data, LD, DAB, and TEM analysis, and naked mole-rat western blot data. Tukey post-tests and Dunnett's multiple comparisons test were used, where appropriate, to assess changes in variables from normoxic/resting conditions. Western blot data from CHH, CHM, CHP, and GC were analyzed by one-sided Welch's $t$-tests. Values are reported as mean ± SEM. All statistical analyses were performed using GraphPad Prism 9 (GraphPad Prism, La Jolla, CA, USA), with a significance level of $p < 0.05$.

**Reporting summary**. Further information on research design is available in the Nature Research Reporting Summary linked to this article.

## Data availability

All data generated in this study have been deposited in the Figshare database under accession codes https://figshare.com/s/e5e7767403970f6285de, https://figshare.com/s/7e40f33306fae3289e0c, https://figshare.com/s/2a77c3c04db76b5a25d1, and https://figshare.com/s/9e1bdce1f4538252c474. Source data are provided with this paper.

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

## Acknowledgements

We would like to thank the University of Ottawa and University of Pretoria animal care and veterinary services teams for their assistance in animal handling and husbandry. We would like to thank Drs. S. Kelly Sears and Jeannie Mui from the Facility for Electron Microscopy Research at McGill University for TEM services and Dr. Ryan Russell, University of Ottawa, for expert advice on electron micrograph analyses. We would also like to thank Dr. Michael Downey and Alix Denoncourt, University of Ottawa, for their assistance with co-immunoprecipitation experiments. This work was supported by an NSERC Discovery grants to MEP (#04229), GJT (#05089), and MEH (#04468), a Canada Research Chair to MEP (#950-230954), and an University of Ottawa Research Chair to MEH. Collection and housing of mole-rats in Africa were funded by a SARChI grant (GUN 64576) to NCB.

## Author contributions

M.E.P., G.J.T., R.S., and M.E.H. conceived of and designed the study. B.v.J. and D.W.H. collected the South African animals with assistance from N.B. and M.E.P; H.C. and R.S. performed the molecular biology experiments. H.C., M.E.H., and A.K. performed the physiology experiments and NM analyzed the physiological data. B.L. analyzed the EM data. Z.E.H. and M.E.H. analyzed the H&E data. M.E.P., R.S., G.T., and Z.E.H. conducted statistical analysis. M.E.P., R.S., and M.E.H. wrote the paper. All authors read and gave final approval of the published version and agree to be accountable for all content therein.

## Competing interests

The authors declare no competing interests.

## Additional information

**Peer review information** *Nature Communications* thanks David Haig, Yihai Cao and the other anonymous reviewer(s) for their contribution to the peer review this work. Peer reviewer reports are available.

