## [Peer Review File. · Nature Communications]

Reviewers' Comments:

Reviewer #1:

Remarks to the Author:

I found the paper very interesting and recommend publication. The authors show that naked mole rats exhibit robust non-shivering thermogenesis under normoxic conditions, contrary to some claims that naked mole rats are poikilothermic. However, naked mole rats rapidly shut-down thermogenesis under hypoxic conditions. A similar shut-down of thermogenesis under hypoxic conditions is observed in other social species of mole rats but not in a solitary species. These observations make intuitive sense. The oxygen consumption of groups is greater than solitary individuals and therefore the reduction of oxygen consumption (consequent on the shut-down of thermogenesis) has aspects of altruistic behavior for the good of the group. [A formal analysis would need to take account of the interaction between group size and body size because smaller bodies consume less oxygen.]

I have one minor difference of interpretation from the authors. At lines 36–37 and line 83, the authors state that thermogenesis is down-regulated in hypoxia to “conserve energy”. This does not seem right. Thermogenesis is down-regulated in hypoxia to conserve oxygen for other metabolic activities (conservation of energy is a by-product).

line 409 “mol-rat”

Reviewer #3:

Remarks to the Author:

Major comments:

The paper by Pamenter et al., reveals that thermogenesis in brown adipose tissue is shut off when naked mole-rats acutely encounter atmospheres containing 7% oxygen for 1 hour. These novel observations in switching off facultative thermogenesis are striking and of interest to thermoregulatory biologists. I am disappointed by the lack of a well articulated, rigorous experimental design and as such I am not so convinced of the mechanism put forward. For a high caliber journal, such as this one, I would expect multiple experiments to be undertaken (rather than the exposure for 1h to 1 level of oxygen), titrating the doses of hypoxia at which thermogenesis is switched off and the provision of a much more rigorous and in-depth molecular assessments. I query the data quality given the unusual methods for protein extraction from fat tissue, coupled with poor quality Ponceau and Western blots that imply lipid likely is contaminating samples. This may be exacerbated post hypoxia when more cellular debris may be present (given the low centrifugation speeds mentioned) . Furthermore, a large component of this study is based upon the use of antibodies. In exotic species this requires the provision of information regarding homology of the epitope (and affinity) especially in multi species comparative studies. Additional confirmatory experiments, using techniques other than Western blots are needed before such sweeping conclusions can be drawn.

Naked mole-rats were only exposed for 1 hour to only 1 level (7% O₂) of hypoxia yet these data are later compared to a study undertaken on several other species (albeit with small n's) subjected for a longer period (3h) and to a different level of hypoxia (5% O₂). I don't think you can make this comparison.

Pamenter and colleagues report that following 1h of exposure to 7% oxygen “decreased and jagged cristae and undefined mitochondrial membranes, consistent with the possibility of mitochondrial degradation”. So rapid a rate of mitochondrial demise is unusual. The authors should investigate whether mitochondrial quality control pathways are activated in naked mole-rats under hypoxia and the rapidity of this response. For example, the authors should investigate core components of the mitophagy machinery in iBAT.

UCP1 positive beige fat cells are also implicated in thermogenesis. Depletion of UCP1 and reduced mitochondria function is known to occur during Beige-to-White Adipocyte Transition. The authors should therefore clearly demonstrate that the main cell type impacted upon hypoxia is BAT and is

not due to a Beige-to-White Adipocyte Transition.

Of major concern in this manuscript is the outlined methodology for protein extraction and quantification. It is well-known that extracting proteins from fat tissue is problematic and requires more rigorous protocols. The choice of lysis buffer and the over simplified methods outlined for extracting soluble proteins seems problematic: For extraction of proteins from fats, the use of "0.25 M sucrose + protease inhibitors" without added detergents and/or multiple centrifugation steps at higher speeds is unlikely to facilitate rigorous extraction of proteins in fat tissue. The authors should explain their methods in more detail and explain how they removed the lipids from their protein extracts. This would be exacerbated after hypoxia where there could be more debris associated with mitochondrial disintegration. It is well known that lipid contamination interfere with Bradford assays for protein determination. Indeed their images suggest this may be a problem here:- Their Ponceau stained images, particularly in Figure 4 suggest that lipids likely have contaminated their samples. The authors claim they loaded 10 ug of protein in each lane, yet it appears that some lanes have 2-5x more protein than others, either reflecting pipetting errors or, more likely, these samples had variable amounts of lipids which have affected their protein quantification. This level of variability in protein quantitation in the same tissue in samples from the same species is not acceptable. Moreover, the Ponceau images are of very poor resolution making quantification extremely difficult. Did the authors titrate different amounts of lysate and quantify the Ponceau intensity? This would have been useful. Given the huge change in UCP levels, I am certain something is changing pre and post hypoxia, however, given the quality of these blots, and the potential problems in the extraction techniques outlined above, I don't believe the huge magnitude they report can possibly be correct; especially if mitochondrial debris, and lipids accumulate post treatment affecting UCP extraction

Pamenter et al. observe an increase in global protein ubiquitylation in iBAT under hypoxia of several but not all the proteins in the blot. The authors should determine whether ubiquitylation is specific to mitochondria proteins (such as UCP1) using, for example, unbiased mass spectrometry approaches.

Along the same lines, is the increase in global protein ubiquitylation specific to iBAT? To support and strengthen the authors conclusions, they should quantify global protein ubiquitylation in white and beige adipose cells and non-adipose tissue upon hypoxia in this species.

Further investigation into the mechanism by which UCP1 is so rapidly and completely decreased in iBAT upon hypoxia is warranted. Previous studies have identified succinylation as a potent regulator of UCP1 in BAT, mediated by Sirt5. The authors should determine whether a similar mechanism occurs in mole-rats.

The lack of change in the various parameters when experiments are undertaken at 36°C is surprising. This temperature is above normal body temperature and above the TNZ; given that animals are heat stressed at this temperature surely the iBAT UCP changes should be more pronounced? Are there changes in chaperones or HIF1a?

The authors claim that "As in naked mole-rats, iBAT homogenates from CHH, CHM, and CHP also exhibited significant decreases or a strong trend towards decreasing UCP1 expression in hypoxia (Fig. 4)". The data presented in Figure 4 does not support this claim and is not consistent with the dramatic change in UCP1 observed in naked mole-rats (Figure 2A). Pamenter and colleagues should revise this data and the conclusions drawn from it and also postulate why UCP1 is not readily degraded upon hypoxia in these closely related species, despite longer periods of hypoxia and a lower % oxygen (5% not 7%).

Cosmetic issues:

The introduction is overly long and off topic, with the authors devoting an entire page to whether or not the animals are poikilothermic. This has nothing to do with the topic under study. Line 54-76 could be shortened to state that naked mole-rats are thermally labile or heterothermic, with some studies using dry air showing that body temperature closely tracks that of ambient and others showing that the temperature differential between ambient and body temperature may

range between 1-13°C (refs). This differential may depend upon the amount of iBAT and prior cold exposure and concomitant endocrine regulation (refs). Then talk about iBAT regulation and thermogenic potential.

The paper is overloaded with multiple self-citations, often providing multiple references for the same fact from the multiple papers the authors have published on this topic. This creates a bad impression, rather the authors should choose the paper that is most appropriate for the fact they want to reference – possibly their 1st paper on the topic and simply cite that.

Although the authors provide information that 73 animals were used in this study; it is not clear how many animals were used in each experiment and if the same animals were used repeatedly, if so what was the time interval between measurements or was each of the 73 animals only used once? How many of these animals were used for tissue harvesting for this study. Please provide clearly provide this information. E.g. how many animals were used for TEM? How many in Western blot studies? How many in each of the hypoxic in vivo assessments. Without this information, I cannot comment on the validity of the statistical analyses.

The partial pressure of oxygen at Pretoria but not Ottawa is provided. For completeness provide both.

For completeness include the data from the recovery after hypoxia at 20°C; the authors provide this information for the other 2 temperatures at which hypoxic assessments were made.

Both RFID measurements of body temperature and thermographic images were undertaken; how do these 2 data sets compare?

Reviewer #4:

Remarks to the Author:

This manuscript describes decreasing BAT activation in naked mole rats under hypoxic conditions. Upon cold exposure, a key strategy for mammals to sustain their core body temperature is to either reduce heat loss by accelerating oxygen consumption and non-shivering thermogenesis. In the case of NMRs, their core body temperatures can decrease along the ambient temperature changes, a strategy to survive without losing unnecessary energy when white adipose tissues are minimal. The authors used an acute hypoxia (7% O₂) exposure in their experimental settings to investigate the BAT-related thermogenesis. Their findings are interesting and physiologically important in understanding BAT thermogenesis. However, this version of manuscript can be improved by considering the following aspects in a revised version:

1) To make this study more physiologically relevant, is it possible to prolong the hypoxia exposure? In a prolonged experimental setting, they would be able to study if adaptation would play any roles. This experiment should be feasible because the NMRs can tolerate to hypoxia for weeks.

2) I do not know how precious these animals are. The authors have not provide any histological results of their study. It would be very important for them to provide immunohistochemical results to support their conclusions. Because most of BAT activation markers including UCP1 and mitochondrial proteins are intracellular proteins, the commercially available antibodies should also work in rats. In fact, some studies using rats as a model already provide supportive evidence.

3) Giving the fact that thermogenesis is an oxygen-dependent energy consumption process, it is quite logic that deprivation of oxygen in the ambient environment would impair non-shivering thermogenesis. Is it possible for the authors to do this same experiments using experimental rats as a control?

4) Fig. 2 only presents results under thermoneutrality (30°C). It would important for the authors to provide results under cold conditions.

5) What about shivering thermogenesis under cold and hypoxic conditions?

6) Do NMRs also have WATs? I assume that they have visceral WAT. If so, what about visceral WAT changes under cold and hypoxic conditions? These changes may contribute to NMR global metabolism and survival. A recent study shows that visceral fat switches to a brown-like phenotype in mice exposed to very low temperatures (Yang et al., JCI Insight. 2017 Feb 23; 2(4): e89044.).

7) They authors should provide non-shivering thermogenic metabolism data using norepinephrine. The metabolic changes are quite important.

8) The authors should consider to cite the following articles:

Yang et al., JCI Insight. 2017 Feb 23; 2(4): e89044

Xue et al., Cell Metab. 2009 Jan 7;9(1):99-109. doi: 10.1016/j.cmet.2008.11.009.

Replies to the first reviewer

Reviewer #1 overall comments: I found the paper very interesting and recommend publication. The authors show that naked mole rats exhibit robust non-shivering thermogenesis under normoxic conditions, contrary to some claims that naked mole rats are poikilothermic. However, naked mole rats rapidly shut-down thermogenesis under hypoxic conditions. A similar shut-down of thermogenesis under hypoxic conditions is observed in other social species of mole rats but not in a solitary species. These observations make intuitive sense. The oxygen consumption of groups is greater than solitary individuals and therefore the reduction of oxygen consumption (consequent on the shut-down of thermogenesis) has aspects of altruistic behavior for the good of the group. [A formal analysis would need to take account of the interaction between group size and body size because smaller bodies consume less oxygen.]

Specific comments:

1. I have one minor difference of interpretation from the authors. At lines 36-37 and line 83, the authors state that thermogenesis is down-regulated in hypoxia to "conserve energy". This does not seem right. Thermogenesis is down-regulated in hypoxia to conserve oxygen for other metabolic activities (conservation of energy is a by-product).

Response: We appreciate the enthusiasm of this reviewer for our study and would be pleased to modify the text as suggested regarding the semantics of our discussions of metabolic energy savings in relations to thermogenesis.

Replies to the third reviewer

Reviewer #3 overall comments: The paper by Pamerter et al., reveals that thermogenesis in brown adipose tissue is shut off when naked mole-rats acutely encounter atmospheres containing 7% oxygen for 1 hour. These novel observations in switching off facultative thermogenesis are striking and of interest to thermoregulatory biologists. I am disappointed by the lack of a well articulated, rigorous experimental design and as such I am not so convinced of the mechanism put forward. For a high caliber journal, such as this one, I would expect multiple experiments to be undertaken (rather than the exposure for 1h to 1 level of oxygen), titrating the doses of hypoxia at which thermogenesis is switched off and the provision of a much more rigorous and in-depth molecular assessments. I query the data quality given the unusual methods for protein extraction from fat tissue, coupled with poor quality Ponceau and Western blots that imply lipid likely is contaminating samples. This may be exacerbated post hypoxia when more cellular debris may be present (given the low centrifugation speeds mentioned) . Furthermore, a large component of this study is based upon the use of antibodies. In exotic species this requires the provision of information regarding homology of the epitope (and affinity) especially in multi species comparative studies. Additional confirmatory experiments, using techniques other than Western blots are needed before such sweeping conclusions can be drawn.

Response: In some of the WB analyses conducted on NMR BAT samples, we did use a positive control sample (mouse BAT mitochondria extracts) to examine if the antibodies work for NMR. In these blots, we showed that UCP1 and OXPHOS proteins in the positive control are at the same molecular weights of proteins detected in NMR blots. Please see below for other responses to your specific comments.

Specific comments:

1. Naked mole-rats were only exposed for 1 hour to only 1 level (7% O₂) of hypoxia yet these data are later compared to a study undertaken on several other species (albeit with small n^os) subjected for a longer period (3h) and to a different level of hypoxia (5% O₂). I don't think you can make this comparison.

Response: We agree that it would be ideal if we had BAT samples from NMR and African mole rats at the same hypoxia time point for proper comparison. Unfortunately, BAT from NMR and African mole rats were collected at different times. Furthermore, BAT samples from African mole rats are unique and rare and it is not feasible to obtain additional samples from these animals. Importantly, a longer exposure to hypoxia (3 hrs) in other African mole-rats did not cause the dramatic changes (decreases in UCP1 and no decreases in OXPHOS) seen in NMRs in just 1 hr. This highlights the uniqueness and rapidity of the NMR response. However, it would be relatively straightforward to repeat our molecular biology experiments in NMRs exposed to 3 hrs of hypoxia to provide a direct and more equal comparison between the species.

2. Pamenter and colleagues report that following 1h of exposure to 7% oxygen "decreased and jagged cristae and undefined mitochondrial membranes, consistent with the possibility of mitochondrial degradation". So rapid a rate of mitochondrial demise is unusual. The authors should investigate whether mitochondrial quality control pathways are activated in naked mole-rats under hypoxia and the rapidity of this response. For example, the authors should investigate core components of the mitophagy machinery in iBAT.

Response: We agree that it is a valid point and we would be able to conduct this additional analysis in a reasonable timeframe.

3. UCP1 positive beige fat cells are also implicated in thermogenesis. Depletion of UCP1 and reduced mitochondria function is known to occur during Beige-to-White Adipocyte Transition. The authors should therefore clearly demonstrate that the main cell type impacted upon hypoxia is BAT and is not due to a Beige-to-White Adipocyte Transition.

Response: BAT samples used in our study were collected from the interscapular region which mainly consists of classic BAT. Beige fat is distinct from classic BAT (Choe *et al.*, 2016) and we have 25 years of experience differentiating between the two forms.

4. Of major concern in this manuscript is the outlined methodology for protein extraction and quantification. It is well-known that extracting proteins from fat tissue is problematic and requires more rigorous protocols. The choice of lysis buffer and the over simplified methods outlined for extracting soluble proteins seems problematic: For extraction of proteins from fats, the use of "0.25 M sucrose + protease inhibitors" without added detergents and/or multiple centrifugation steps at higher speeds is unlikely to facilitate rigorous extraction of proteins in fat tissue. The authors should explain their methods in more detail and explain how they removed the lipids from their protein extracts. This would be exacerbated after hypoxia where there could be more debris associated with mitochondrial disintegration. It is well known that lipid contamination interfere with Bradford assays for protein determination. Indeed their images suggest this may be a problem here:- Their Ponceau stained images, particularly in Figure 4 suggest that lipids likely have contaminated their samples. The authors claim they loaded 10 ug of protein in each lane, yet it appears that some lanes have 2-5x more protein than others, either reflecting pipetting errors or, more likely, these samples had variable amounts of lipids which have affected their protein quantification. This level of variability in protein quantitation in the same tissue in samples from the same species is not acceptable. Moreover, the Ponceau images are of very poor resolution making quantification extremely difficult. Did the authors titrate different amounts of lysate and quantify the Ponceau intensity? This would have been useful. Given the huge change in UCP levels, I am certain something is changing pre and post hypoxia, however, given the quality of these blots, and the potential problems in the extraction techniques outlined above, I don't believe the huge magnitude they report can possibly be

correct; especially if mitochondrial debris, and lipids accumulate post treatment affecting UCP extraction

Response: In our study we used a standard and simple protocol to prepare BAT homogenates for WB analyses. BAT is a very soft but proteinaceous tissue that can be homogenized easily and when the homogenates are sonicated, that is enough to release proteins from BAT homogenates. In our study, we are not isolating BAT mitochondria and for that reasons we do not have to use that many centrifugation steps. Regarding the reviewer's comments on lipid contaminants in the BAT homogenate samples, we first spun the BAT homogenate samples at 8500g in order to get rid out of most of the lipids followed by a washing spin to get rid of any residual amount of lipids. Importantly, protease inhibitors were added to the lysis buffer to prevent any protein degradation and all the steps were conducted on ice.

5. Pamerter et al. observe an increase in global protein ubiquitylation in iBAT under hypoxia of several but not all the proteins in the blot. The authors should determine whether ubiquitylation is specific to mitochondria proteins (such as UCP1) using, for example, unbiased mass spectrometry approaches.

Response: We thank the reviewer for this comment and agree that the evaluation of the level of ubiquitylation of UCP1 and other mitochondria proteins before and after hypoxia using immunoprecipitation or mass spectrometry approaches would be useful. However, our initial interest was to measure any changes in the total level of ubiquitylation in BAT which can be used as a comprehensive indicator of potential protein degradation mechanisms. For example, two well-known mechanisms involved in mitochondrial protein degradation are the ubiquitination proteasome system (UPS) and the mitophagy system (Harper et al., 2018; Lavie et al., 2018; Pickles et al., 2018). Both mechanisms facilitate the turnover of proteins by ubiquitinating proteins that are targeted for degradation. For that reason, we first probed for the total ubiquitylation level.

6. Along the same lines, is the increase in global protein ubiquitylation specific to iBAT? To support and strengthen the authors conclusions, they should quantify global protein ubiquitylation in white and beige adipose cells and non-adipose tissue upon hypoxia in this species.

Response: We thank the reviewer for mentioning this point. We agree it would be interesting to look at changes in the total level of protein ubiquitylation in white and beige fats during hypoxia. We would be pleased to include this data in a revised manuscript.

7. Further investigation into the mechanism by which UCP1 is so rapidly and completely decreased in iBAT upon hypoxia is warranted. Previous studies have identified succinylation

as a potent regulator of UCP1 in BAT, mediated by Sirt5. The authors should determine whether a similar mechanism occurs in mole-rats.

Response: We thank the reviewer for this great point. New advances in the field of BAT study show how post-translation modifications can control BAT thermogenesis including sulfenylation and succinylation of UCP1 (Chouchani et al., 2016; Wang et al., 2019). However, in the present study, we are more interested in looking at the effect of hypoxia at the protein level.

8. The lack of change in the various parameters when experiments are undertaken at 36°C is surprising. This temperature is above normal body temperature and above the TNZ; given that animals are heat stressed at this temperature surely the iBAT UCP changes should be more pronounced? Are there changes in chaperones or HIF1a?

Response: We would not expect to see any changes in UCP1 function or BAT thermogenesis in NMRs housed above their thermoneutrality because iBAT is not active and there is no need for thermoregulation at such high temperature.

9. The authors claim that “As in naked mole-rats, iBAT homogenates from CHH, CHM, and CHP also exhibited significant decreases or a strong trend towards decreasing UCP1 expression in hypoxia (Fig. 4)”. The data presented in Figure 4 does not support this claim and is not consistent with the dramatic change in UCP1 observed in naked mole-rats (Figure 2A). Pamenter and colleagues should revise this data and the conclusions drawn from it and also postulate why UCP1 is not readily degraded upon hypoxia in these closely related species, despite longer periods of hypoxia and a lower % oxygen (5% not 7%).

Response: We agree with the reviewer that we do not see that very dramatic decreases in the level of UCP1 in CHH, CHM, and CHP. However, there are a significant decrease or a trend of decreases in UCP1 after normalizing the densitometry values of UCP1 bands to the Ponceau staining and running statistical comparisons.

Cosmetic issues:

10. The introduction is overly long and off topic, with the authors devoting an entire page to whether or not the animals are poikilothermic. This has nothing to do with the topic under study. Line 54-76 could be shortened to state that naked mole-rats are thermally labile or heterothermic, with some studies using dry air showing that body temperature closely tracks that of ambient and others showing that the temperature differential between ambient and body temperature may range between 1-13°C (refs). This differential may depend upon the amount of iBAT and prior cold exposure and concomitant endocrine regulation (refs). Then talk about iBAT regulation and thermogenic potential.

Response: We would be pleased to modify our Introduction section to improve the clarity of our message and better set up data presentation.

11. The paper is overloaded with multiple self-citations, often providing multiple references for the same fact from the multiple papers the authors have published on this topic. This creates a bad impression, rather the authors should choose the paper that is most appropriate for the fact they want to reference – possibly their 1st paper on the topic and simply cite that.

Response: Presumably the reviewer is referring to our initial discussion of NMR responses to hypoxia in the introduction, which is largely based on work out of the Pamerter lab because this is the only lab currently exploring physiological responses to hypoxia in this species. However, we would be pleased to streamline the number of citations to our work here and throughout the paper.

12. Although the authors provide information that 73 animals were used in this study; it is not clear how many animals were used in each experiment and if the same animals were used repeatedly, if so what was the time interval between measurements or was each of the 73 animals only used once? How many of these animals were used for tissue harvesting for this study. Please provide clearly provide this information. E.g. how many animals were used for TEM? How many in Western blot studies? How many in each of the hypoxic in vivo assessments. Without this information, I cannot comment on the validity of the statistical analyses.

Response: The sample sizes for each individual experiment were already included in the figure legends of our initial submission; however, we would be pleased to repeat this information in the body of the manuscript as well. None of the animals were used repeatedly.

13. The partial pressure of oxygen at Pretoria but not Ottawa is provided. For completeness provide both.

Response: The partial pressure of O₂ in Ottawa is ~ 21kPa and we will be pleased to include this in our revised manuscript.

14. For completeness include the data from the recovery after hypoxia at 20oC; the authors provide this information for the other 2 temperatures at which hypoxic assessments were made.

Response: We are unsure as to what the reviewer is referring. Recovery data after hypoxia was already included for our 20C experiments in our initial submission.

15. Both RFID measurements of body temperature and thermographic images were undertaken; how do these 2 data sets compare?

Response: We are unsure as to what the reviewer is referring. These data are included in each of our temperature graphs and are readily comparable in Figs 1, S1, and S2.

Replies to the fourth reviewer

Reviewer #4 overall comments: This manuscript describes decreasing BAT activation in naked mole rats under hypoxic conditions. Upon cold exposure, a key strategy for mammals to sustain their core body temperature is to either reduce heat loss by accelerating oxygen consumption and non-shivering thermogenesis. In the case of NMRs, their core body temperatures can decrease along the ambient temperature changes, a strategy to survive without losing unnecessary energy when white adipose tissues are minimal. The authors used an acute hypoxia (7% O₂) exposure in their experimental settings to investigate the BAT-related thermogenesis. Their findings are interesting and physiologically important in understanding BAT thermogenesis. However, this version of manuscript can be improved by considering the following aspects in a revised version:

1. To make this study more physiologically relevant, is it possible to prolong the hypoxia exposure? In a prolonged experimental setting, they would be able to study if adaptation would play any roles. This experiment should be feasible because the NMRs can tolerate to hypoxia for weeks.

Response: Recent measurements of NMR burrows suggest that they are not particularly hypoxic (Holtze et al., 2018); however, these animals putatively experience severe hypoxia in their nest chambers. Nonetheless, most colony members would only spend a few hours at a time in this chamber and so it is believed that NMRs would never experience chronic hypoxia of the duration proposed by the reviewer. We agree that these experiments would be interesting but they would not be ecophysiologically relevant and so the resulting data would be somewhat meaningless in this context.

2. I do not know how precious these animals are. The authors have not provided any histological results of their study. It would be very important for them to provide immunohistochemical results to support their conclusions. Because most of BAT activation markers including UCP1 and mitochondrial proteins are intracellular proteins, the commercially available antibodies should also work in rats. In fact, some studies using rats as a model already provide supportive evidence.

Response: We would be pleased to conduct the experiments suggested by the reviewer and would be able to complete these in a reasonable timeframe.

3. Given the fact that thermogenesis is an oxygen-dependent energy consumption process, it is quite logical that deprivation of oxygen in the ambient environment would impair non-shivering thermogenesis. Is it possible for the authors to do these same experiments using experimental rats as a control?

Response: There is considerable data available in the literature exploring the effects of cold and hypoxia on thermogenesis in hypoxia-intolerant rodents. We would be pleased to discuss this literature at greater length in a revised manuscript.

4. Fig. 2 only presents results under thermoneutrality (30°C). It would be important for the authors to provide results under cold conditions.

Response: This data was already included in our supplementary data demonstrating thermal responses to an ambient temperature of 20°C (Fig. S1).

5. What about shivering thermogenesis under cold and hypoxic conditions?

Response: This information was already included in our supplementary data (Fig. S1).

6. Do NMRs also have WATs? I assume that they have visceral WAT. If so, what about visceral WAT changes under cold and hypoxic conditions? These changes may contribute to NMR global metabolism and survival. A recent study shows that visceral fat switches to a brown-like phenotype in mice exposed to very low temperatures (Yang et al., JCI Insight. 2017 Feb 23; 2(4): e89044.).

Response: Yes, NMRs do express WAT but the study cited by the reviewer exposed mice to several weeks of lower temperature to induce fat switching. It is notable that our exposures were acute and we measured responses that happened within hours. This is too fast for WAT conversion to BAT and so is unlikely to play a role here. This would be an interesting question to explore in a future study.

7. The authors should provide non-shivering thermogenic metabolism data using norepinephrine. The metabolic changes are quite important.

Response: This data was already provided in our supplementary materials (Fig. S2).

References:

1. Choe, S.S., Huh, J.Y., Hwang, I.J., Kim, J.I., and Kim, J.B. (2016). Adipose Tissue Remodeling: Its Role in Energy Metabolism and Metabolic Disorders. *Front. Endocrinol.* 7.
2. Chouchani, E.T., Kazak, L., Jedrychowski, M.P., Lu, G.Z., Erickson, B.K., Szpyt, J., Pierce, K.A., Laznik-Bogoslavski, D., Vetrivelan, R., Clish, C.B., et al. (2016). Mitochondrial ROS regulate thermogenic energy expenditure and sulfenylation of UCP1. *Nature.* 532, 112–116.
3. Harper, J.W., Ordureau, A., and Heo, J.-M. (2018). Building and decoding ubiquitin chains for mitophagy. *Nature Reviews Molecular Cell Biology.* 19, 93–108.
4. Holtze, S., Braude, S., Lemma, A., Koch, R., Morhart, M., Szafranski, K., Platzer, M., Alemayehu, A., Goeritz, F., and Hildebrandt, T.B. (2018). The microenvironment of naked mole-rat burrows in East Africa. *African Journal of Ecology* 56, 279-289.
5. Lavie, J., De Belvalet, H., Sonon, S., Ion, A.M., Dumon, E., Melser, S., Lacombe, D., Dupuy, J.-W., Lalou, C., and Bernard, G. (2018). Ubiquitin-Dependent Degradation of Mitochondrial Proteins Regulates Energy Metabolism. *Cell Reports.* 23, 2852–2863.
6. Pickles, S., Vigi, P., and Youle, R.J. (2018). Mitophagy and Quality Control Mechanisms in Mitochondrial Maintenance. *Current Biology.* 28, R170–R185.
7. Wang, G., Meyer, J.G., Cai, W., Softic, S., Li, M.E., Verdin, E., Newgard, C., Schilling, B., and Kahn, C.R. (2019). Regulation of UCP1 and Mitochondrial Metabolism in Brown Adipose Tissue by Reversible Succinylation. *Molecular Cell.* 74, 844-857.e7.

Reviewers' Comments:

Reviewer #1:

Remarks to the Author:

This is a good study deserving of publication.

Reviewer #3:

Remarks to the Author:

While the authors of this ms have responded to many of the comments by the reviewers and are to be commended for doing so, several concerns remaining the quality of the paper remain. The quality for the Western blots is much improved, however there are data without loading controls and some of these interpretations need to be clarified.

Minor points:

Introduction

Page 3 line 286-287: "Indeed, naked mole-rats expend considerable energy to thermoregulate outside of their thermoneutral zone"; Buffenstein and Yahav (1991, J thermal biol 16:272) reported that metabolic rate at 27-28°C was as high as that of a mouse at 5°C; including this strengthens this sentence and this key thermoregulatory reference should be included in your bibliography.

Page 3 line 287-289 " several studies have reported their ability to maintain Tb well above Ta in a range of temperatures (Tb-Ta differential ranging from 0.0 to >13.0°C in animals exposed to Tas ranging from 37-10°C)18-20" The information provided in parentheses is contradictory to your point. You mention that animals are able to maintain Tb well above Ta, yet state Tb-Ta ranges from 0.0 to >13°C.... A differential of zero suggests that in some of these studies Tb is exactly the same as Ta! Please clarify and also note these differentials depend to a large extent on the experimental conditions and duration of exposures, since the references cited use very different experimental protocols including multiple animal housing and provision of nesting material that will influence the Tb-Ta differential.

Page 4 line 350 the 1st study to report functional BAT through the induction of adrenergic stimulation of non-shivering thermogenesis was Hislop and Buffenstein 1994 (J Thermal Bio 19;25-32). There are several other seminal studies (outside of the Pamenter group) directly pertinent to this ms that are not referenced.

Methods

Page 8 line 438-440. "The 36°C experimental temperature was selected to examine thermogenic responses in an Ta at which naked mole-rats would likely not need to actively thermoregulate (i.e., above their thermoneutral zone)." As stated, this temperature is above the TNZ (30-34), so metabolic rate will be higher than within the TNZ and likely the animals will be subjected to a mild heat stress. Could this impact your findings?

Page 8 line 447 Since you are mentioning the partial pressure of oxygen in Pretoria in kPa, why have you responded to a reviewers request to provide the same information for Ottawa by giving the % atmospheric oxygen and not in kPa?

Results

Fig 1. Can you explain why core body temperature (usually the highest Tb one measures) is lower than that of the rump? Also, can you explain how Tb-Ta is less than 0 when Tb is measured (fig 1D); I presume this is because of measurements undertaken in dry air so that the vapor pressure gradient for evaporative water loss leads to greater heat loss than metabolic heat production ; but do the authors have an explanation for this unusual observation that Tb-Ta is far greater than Trump-Ta during hypoxia? These data are hard to reconcile.

Fig 2A. UCP levels after 1h hypoxia appear to fall into 2 distinct clusters in which 5 animals have no measureable UCP (rapid degradation?) and the other 5 animals have UCP levels in keeping with

normoxia (no degradation?). Can you explain how ½ the animals in this experimental cohort have no detectable levels within 60 min of hypoxia. Also why are the n's in each of these bars for UCP1 and UCP3 different. Surely the same lysates were used to measure both UCP1/UCP3. Clearly the 2 blots shown were run at different times, where are the loading controls for these and other western blots (Fig 2-6)?

Discussion

Page 26 line 522-523 I disagree with the statement 'Furthermore, the absence of any change in Fis1 expression at 1 hr suggests that hypoxia fission of healthy mitochondria occurs rapidly during acute hypoxia'. FIS is responsible for peripheral mitochondrial fission and leads to subsequent mitochondrial degradation whereas MFF is upregulated when mitochondria undergo midzone fission to generate more 'healthy' mitochondria (Kleele et al 2021). Based on Fig 5, a robust increase in DRP1 S616, the slight but insignificant increase in FIS and an unchanged MFF level upon 1 h exposure to hypoxia suggest mitochondria were undergoing a low degree of degradation as opposed to what was suggested by the authors. Perhaps the authors can address this alternate explanation.

Reviewer #4:

Remarks to the Author:

The authors have satisfactorily addressed my previous concerns. Congratulations on contributing to this important piece of work.

Replies to the first reviewer

Reviewer #1 overall comment: This is a good study deserving of publication.

Response: Thank you again for your time and effort in reviewing our submission!

Replies to the third reviewer

We would like to take this opportunity to thank you for your kind and helpful comments that have helped us to substantially improve our submission. All comments and suggestions have been addressed as detailed below:

Reviewer #3 overall comments: While the authors of this ms have responded to many of the comments by the reviewers and are to be commended for doing so, several concerns remaining the quality of the paper remain. The quality for the Western blots is much improved, however there are data without loading controls and some of these interpretations need to be clarified.

Minor comments:

1. Introduction Page 3 line 286-287: “Indeed, naked mole-rats expend considerable energy to thermoregulate outside of their thermoneutral zone”; Buffenstein and Yahav (1991, J thermal biol 16:272) reported that metabolic rate at 27-28°C was as high as that of a mouse at 5°C; including this strengthens this sentence and this key thermoregulatory reference should be included in your bibliography.

Response: Thank you for this suggestion. We have included this reference and a discussion of this finding in our revised introduction.

2. Page 3 line 287-289 “several studies have reported their ability to maintain T_b well above T_a in a range of temperatures ($T_b - T_a$ differential ranging from 0.0 to $>13.0^\circ\text{C}$ in animals exposed to T_a s ranging from $37-10^\circ\text{C}$)18-20” The information provided in parentheses is contradictory to your point. You mention that animals are able to maintain T_b well above T_a , yet state $T_b - T_a$ ranges from 0.0 to $>13.0^\circ\text{C}$ A differential of zero suggests that in some of these studies T_b is exactly the same as T_a ! Please clarify and also note these differentials depend to a large extent on the experimental conditions and duration of exposures, since the references cited use very different experimental protocols including multiple animal housing and provision of nesting material that will influence the $T_b - T_a$ differential.

Response: Please note that our comment here references experimental data that includes measurements from animals held in T_a 's above the TNZ of this species (i.e., up to 37°C in some papers). When held in the upper range of their TNZ and above, several studies have reported that the NMR T_b is approximately the same as T_a . Given that most temperature sensors have some degree of error that typically approaches $\pm 1^\circ\text{C}$ (see below), it is not particularly surprising that measurements where $T_b = T_a$ (approximately) have been reported for this species.

However, at increasingly colder temperatures, the separation between T_b and T_a increases (along with increased VO_2) in most experimental conditions. Nonetheless, we agree that this sentence was written in a confusing manner and have revised it to focus upon the

upper end of the Ta-Tb ranges used in previous studies, which is the focus of our discussion at this stage of the text.

3. Page 4 line 350 the 1st study to report functional BAT through the induction of adrenergic stimulation of non-shivering thermogenesis was Hislop and Buffenstein 1994 (J Thermal Bio 19;25-32). There are several other seminal studies (outside of the Pamenter group) directly pertinent to this ms that are not referenced.

Response: We agree that this reference is important and have reinserted it here. Please note that we included other seminal studies of BAT and thermogenic capacity in NMRs, in our initial submission. However, in response to your comment in the first round of revisions that we had too many citations for each point in the introduction, we deleted these references and only included a reference to the most comprehensive study of BAT function in NMRs. However, we are happy to reinsert excellent early references. Please let us know if there are additional key references that you would like for us to reinsert.

4. Methods Page 8 line 438-440. “The 36°C experimental temperature was selected to examine thermogenic responses in an Ta at which naked mole-rats would likely not need to actively thermoregulate (i.e., above their thermoneutral zone).” As stated, this temperature is above the TNZ (30-34), so metabolic rate will be higher than within the TNZ and likely the animals will be subjected to a mild heat stress. Could this impact your findings?

Response: We agree that NMRs would need to thermoregulate (but not engage thermogenesis) at 36C. However, the issue here is related to poor phrasing on our part. Our intent was to state that we chose this temperature because it was above the temperature at which NMRs would need to actively generate heat, rather than it being a temperature at which they would not need to thermoregulate. Since our study focused on the function of BAT, we thus used this treatment as a negative control to test animals in a condition in which BAT was expected to not be strongly activated. We have revised this sentence to refer explicitly to the generation of heat, instead of thermoregulation, in the warmer condition.

It is possible that animals treated in 36C were heat stressed, but this seems unlikely. This temperature is within 2C of the TNZ for this species, is generally below stressful Ta levels for other small rodents (all of whom also have fur), and is well within the recorded range for NMR burrows, suggesting that they experience such temperatures for some period of time regularly in nature. In either case, we do not think heat stress (if it occurred) impacted our experiments as stress would not likely alter BAT function significantly within the timeframe of our experiments, and the range of potential change (the scope between Ta and Tb) is very small in these conditions in normoxia, providing little scope for BAT function to impact Tb in hypoxia in this high ambient temperature. In addition, this experimental condition was examined as a control in physiological experiments but was not used in any of our molecular or imaging work. Thus, we do not think that ambient temperatures had any effect on the core mechanistic story detailed in our resubmission.

5. Page 8 line 447 Since you are mentioning the partial pressure of oxygen in Pretoria in kPa, why have you responded to a reviewers request to provide the same information for Ottawa by giving the % atmospheric oxygen and not in kPa?

Response: This oversight has been corrected.

6. Results Fig 1. Can you explain why core body temperature (usually the highest T_b one measures) is lower than that of the rump? Also, can you explain how $T_b - T_a$ is less than 0 when T_b is measured (fig 1D); I presume this is because of measurements undertaken in dry air so that the vapor pressure gradient for evaporative water loss leads to greater heat loss than metabolic heat production ; but do the authors have an explanation for this unusual observation that $T_b - T_a$ is far greater than $T_{rump} - T_a$ during hypoxia? These data are hard to reconcile.

Response: Although measurements of T_{rump} are greater than measurements of T_b in several instances, these differences are relatively small and are not statistically significant within any dataset. We suspect that these temperatures are actually quite similar since the NMR is unable to effectively retain heat and so has a minimal thermal gradient across its body when in hypoxia and not generating heat actively.

It is most likely that the small differences between these two physiological temperature datasets are due to differences in the measurement approach used because T_{rump} was measured using FLIR and T_b was measured using implanted RFID chips. In most cases, the difference between T_{rump} and T_b values are $< 1^\circ\text{C}$, which is within the error for the RFID chips that we used. There is also some degree of uncertainty in the FLIR image analysis. The specific camera we used was factory calibrated and compared recently against a robust blackbody radiation source, but FLIR rates its accuracy at $\pm 1^\circ\text{C}$ (see Playa-Montmany and Tattersall, 2020. *Methods in Ecology and Evolution*, <https://doi.org/10.1111/2041-210X.13563>, for a detailed breakdown of inter-camera accuracy and thermal camera accuracy issues in general). Further complicating matters for the thermal image estimates is that a transparent glass was used to keep the NMR exposed to hypoxic conditions. Although rated at 95% transmissive to infrared, the algorithm FLIR uses to estimate temperature is an empirical, approximate algorithm, and there might be contaminating reflected radiation within the image space that simply cannot be controlled for. Together, the propagation of these errors likely explains the small differences between these temperature datasets.

Similar issues (addition of small errors associated with the measurement of temperature) may explain how $T_b - T_a < 0$ in some datasets. It is notable that across all datasets, individual datapoints for which $T_b - T_a$ is “below” 0 are all within 1°C of zero (with 2 exceptions, which are within 1.5°C). This is again within the combined error expected for our experimental equipment. It is likely that the relationship between T_b and T_a is very close to 0 in these animals and treatment conditions (30C experiments in hypoxia and 36C experiments in both normoxia and hypoxia), given that $T_b - T_a$ for all of the experimental replicates cluster within $\sim \pm 0.5^\circ\text{C}$ of zero in this dataset.

7. Fig 2A. UCP levels after 1h hypoxia appear to fall into 2 distinct clusters in which 5 animals have no measurable UCP (rapid degradation?) and the other 5 animals have UCP levels in keeping with normoxia (no degradation?). Can you explain how ½ the animals in this experimental cohort have no detectable levels within 60 min of hypoxia. Also why are the n's in each of these bars for UCP1 and UCP3 different. Surely the same lysates were used to measure both UCP1/UCP3. Clearly the 2 blots shown were run at different times, where are the loading controls for these and other western blots (Fig 2-6)?

Response: We agree that there is a high degree of variability in several of our datasets. A high degree of intra-animal variability is expected in the study of naked mole-rats because, although our animals are lab-raised, they are not genetic clones (like most lab-mice lines) and thus have considerably more genetic and physiological diversity. In addition, animals were randomly selected from various colonies, within which they likely had differing statuses and thus divergent access to resources, huddling priority, etc.. Finally, the animals we used were not controlled for sex or age and so likely have more variability in their physiology. The other mole-rat species were wild-caught and so we are also unable to control for previous sexual history and development, age, comprehensive health status, and other important variables in these species. For these reasons, we have included an unusually high number of replicates in our analysis of many of the proteins of interest. The UCP1 dataset is a good example of this approach, wherein we have analyzed a very large number of samples to convince ourselves, and thus hopefully readers of our submission, of the significance and validity of our findings related to this protein.

Since the change in UCP1 expression is central to the mechanism outlined in our study, we examined a particularly large number of samples for this protein, including 19 normoxic replicates, 11 replicates from 1 hr hypoxia samples, and 6 replicates for the 3 hr hypoxia treatment. There is some degree of clustering in the 1 hr hypoxic dataset, with some samples having nearly undetectable levels of expression and others higher levels. However, it is important to remember that all these replicates are unpaired...i.e., samples were taken from animals treated in normoxia or hypoxia, and not from the same animals treated in both conditions. Animals in the 1 hr hypoxia group with UCP1 expression levels near the normoxic mean may well have had expression 50% higher in normoxia (as do many samples in the control group).

It is also notable that although several of the 1 hr hypoxia replicates cluster near the normoxic mean, all are below this mean. Conversely, 10/19 of the normoxic samples are greater than all of the 1 hr hypoxia samples, with only 1/19 normoxic replicates having an expression level lower than the 1 hr hypoxia mean. Furthermore, the statistical analysis of this dataset is robust and clear and power analysis indicates that we have sufficient replicates to support our conclusions. Therefore, we are not concerned with the variability in our data.

UCP1 and UCP3 were not analyzed on the same blot or necessarily using the same lysates. For most proteins analyzed, lysates were selected from 6 out of 12 treatment replicates from each treatment condition (plus additional normoxic controls for UCP1). In this fashion, replicate use from our sample pool was randomized for each gel.

Sample loading controls were included in the supplemental information section. Please refer to this section (Fig. S5) to find Ponceau S stain images and full gel scan images from all protein analysis.

8. Discussion Page 26 line 522-523 I disagree with the statement 'Furthermore, the absence of any change in Fis1 expression at 1 hr suggests that hypoxia fission of healthy mitochondria occurs rapidly during acute hypoxia'. FIS is responsible for peripheral mitochondrial fission and leads to subsequent mitochondrial degradation whereas MFF is upregulated when mitochondria undergo midzone fission to generate more 'healthy' mitochondria (Kleele et al 2021). Based on Fig 5, a robust increase in DRP1 S616, the slight but insignificant increase in FIS and an unchanged MFF level upon 1 h exposure to hypoxia suggest mitochondria were undergoing a low degree of degradation as opposed to what was suggested by the authors. Perhaps the authors can address this alternate explanation.

Response: The changes in FIS1 protein (~ 1.19-fold increase, $p = 0.6283$) and MFF (1.09-fold increase, $p = 0.8153$) are very similar and neither approach significance, so it seems unreasonable to dismiss the lack of changes of MFF but extrapolate a non-existent change in FIS1. However, this is likely a discussion of degrees since FIS1 is upregulated at 3 hrs, and thus it is likely that in some cells it is beginning to be upregulated at different time points up to and beyond the 3 hr time point. We have included a comment on this progression in our revised discussion section.

Replies to the fourth reviewer

Reviewer #4 overall comments: The authors have satisfactorily addressed my previous concerns. Congratulations on contributing to this important piece of work.

Response: Thank you again for your time and effort in reviewing our submission!

Reviewers' Comments:

Reviewer #3:

Remarks to the Author:

The authors have adequately addressed most of my concerns.